# Research on Spatial Correlation Characteristics and Their Spatial Spillover Effect of Local Government Debt Risks in China

**Xing Li [1],\*** , **Xiangyu Ge [2],\*** , **Wei Fan [1]** and **Hao Zheng [2]**

1   Economics and Management School, Wuhan University, Wuhan 430072, China; 2017101050057@whu.edu.cn
2   School of Statistics and Mathematics, Zhongnan University of Economics and Law, Wuhan 430073, China; zhenghao@stu.zuel.edu.cn
\*   Correspondence: 200630040104@whu.edu.cn (X.L.); xiangyu_ge@163.com (X.G.)

**Abstract:** Scholars have proposed a series of methods, such as "sustainability of local government debt", to measure local government debt risks. However, these methods have caused a lot of controversy. Based on a macro balance sheet, this study uses an improved "distance to distress" to measure China's local government debt risks and applies a social network model to identify the spatial correlation characteristics, as well as the spillover effect. The results are as follows: (1) The data show multiple and heterogeneous spatial correlations for China's local government debt risks; (2) there are some similarities between the subgroups and seven major geographic regions in China. The links among subgroups are randomly distributed and external; (3) the data manifest a "small world", with a decreasing transitivity since 2014; (4) between these two significant factors, the positive impact of local government competition is more obvious than the division of powers and responsibilities; and (5) the spatial spillover effect of China's local government debt risks results from the combination of local government competition, the division of powers and responsibilities, and local government intervention. This paper provides a scientific basis for obtaining a deeper understanding of China's local government debt risks, and puts forward policy recommendations to strengthen China's debt management.

**Keywords:** local government; debt risks; distance to distress; spatial correlation; spillover effect

## 1. Introduction

Nowadays, the slowdown of economic growth of China is accompanied by a further large-scale expansion of local government debt. According to data from China's Ministry of Finance, at the end of 2018, the debt balance of China's local governments had reached 18,386.2 billion RMB—almost 80 times greater than the value recorded in 1997. Obviously, the expansion of local government debt has exceeded the capacity of the local economy and has attracted worldwide concern [1]. Except for the explicit debt represented by local government bonds, some scholars have pointed out that if implicit debt is included, the average debt ratio of China's local governments had reached 72.3% in 2018, far beyond the international warning level [2]. Additionally, the local government debt risks are roiling more and more of China's provinces [3]. Not surprisingly, the study of contagion of local government debt risks has been reported by Easterly et al. [4]. Under the severe situation in China, the local government debt risks among different provinces have been subjected to all-round monitoring in China. If this monitoring is not conducted, the debts are likely to have an effect on China's financial system, or even the stable operation of the global economy.

Moreover, the local government debt risk is not limited to China, but also extends worldwide, ranging from federal countries such as Spain, America, Mexico, and Brazil, to single countries, such as Italy, Vietnam, and China. As a result, many countries and

international organizations are actively trying to resolve the local government debt risks. For example, Italy has issued the "No. 89 Amendment on Emergency Measures for Competitiveness and Social Justice", Brazil has launched the "Fiscal Responsibility Law", Mexico has announced the "Federal Fiscal Responsibility Act", America has proposed the Ohio model, and the Worldwide Bank Group has released the "World Governance Indicators". Therefore, this research sheds new light on local government debt risks from China's experience for other countries facing similar problems. Recently, there has been a tendency to study the local government debt risks among scholars around the world, and more and more attention has been paid to their measurement and contagion. To be specific, a series of methods, such as "sustainability of local government debt", have been proposed to measure the risks [5–9]. Additionally, spatial econometric models have been applied to analyze the contagion, i.e., the spatial spillover effect [10,11]. However, controversy has arisen over these methods within academia. Specifically, spatial econometric models are mainly required when it comes to studying the scale of local government debt. However, further questions need to be explored. For example, how can local government debt risks be measured more objectively and scientifically? What macro factors can be used to explain the spatial spillover effect of local government debt risks in China? In addition to spatial econometric models, what other empirical model can be used to analyze the spatial spillover effect of local government debt risks?

The foci of this paper are as follows. Firstly, taking China as a sample, data are derived from the Wind database, Qianzhan database, Hexun website, East-Money Choice terminal, and various statistical yearbooks of China to study the local government debt risks, and policy recommendations are provided to strengthen debt management, which can encourage other countries facing similar risks to learn from China's experience. Secondly, more details on the compilation of macro balance sheets of China's local governments can be found in the calculation of "distance to distress", which will shed new light on measuring local government debt risks. Thirdly, it provides a new perspective for analyzing the spatial correlation characteristics and spillover effect of local government debt risks in China, including the local government competition, division of powers and responsibilities, and local government intervention. Fourthly, the social network is successfully introduced to study local government debt risks, which provides another useful and empirical analysis idea for scholars. However, because of the small data size, it is necessary to follow the changes in Chinese local government debt risk and update the sample data in future research.

The rest of the paper is organized as follows. In Section 2, a survey of the literature is provided through a comparison with previous studies. In Section 3, a series of research hypotheses are designed. Based on the macro balance sheet of local governments, the objective estimation of the local government debt risks of each province in China is linked to Section 4. In Section 5, the social network model is applied to identify the spatial correlation characteristics of local government debt risks in China and analyze the macro factors that influence the spatial centrality, as well as the spillover effect of local government debt risks. Naturally, a robust test is considered in Section 6, followed by conclusions with policy recommendations in Section 7.

## 2. Literature Review

### 2.1. Measurement

The measurement of local government debt risks has been one of the research hotspots since the 1970s, and a series of methods have been proposed by scholars. To sum up, these methods can be classified as the "scale of local government debt" substituting for local government debt risks, the "sustainability of local government debt", the "distance to distress (probability of default) of local government debt", and the "risk index system of local government debt", each of which has its own variety of benefits and pitfalls.

It has been widely agreed by many scholars to take the "scale of local government debt" as a substitute for local government debt risks, and the same goes for "the expansion of local

government debt is equivalent to the increase of local government debt risks" [12,13], as well as "government debt issuance should control the scale" [12,14,15]. Accompanied by in-depth study, a widespread discussion has been provoked regarding the view "the expansion of local government debt is equivalent to the increase of local government debt risks". Taking China as an example, which has many large provinces issuing local government bonds and urban investment bonds, including Jiangsu, Zhejiang, and Shanghai, it is interesting to consider whether these provinces are similar to provinces with the most serious risks of local government debt. The answer is no on the basis of relevant papers and research [16,17]. According to one example in China, it is unreasonable to casually equate the expansion of local government debt with an increase of local government debt risks. Generally speaking, when it comes to the scale of local government debt, the "asset capacity" of local governments is evident—the capacity of assets to face liabilities.

The "sustainability of local government debt" is considered one of the most frequently considered approaches by scholars, with roots in the discussion of "fiscal sustainability". An early discussion on fiscal risk is required to measure the local government debt risk, which is the manifestation and destination of fiscal risk. Consequently, studies on "fiscal sustainability" have substantially increased in number, focusing on the topics of fiscal revenue, fiscal expenditure, and government debt. Detailed examinations of the "sustainability of government debt" have been undertaken by Brady and Magazzino [5–9], who have respectively examined the relationship between government primary deficit and debt for 28 European Union countries, G-7 countries, 19 European Monetary Union countries, and Italy, finding a clear co-integrating relationship between government debts and primary deficits. Fundamentally, previous studies have provided enlightenment in terms of the measurement of local government debt risks, but there are still some problems in the application, especially for China. Firstly, the risks of government primary deficits and debt are not exactly the same in a strict sense. To be specific, the risk of government debt focuses on the possibility of government debt repayment, while the risk of primary deficit refers to the ability to fulfill public responsibility when the government's expenditure exceeds the revenue. In terms of judging the government's debt repayment ability on the basis of fiscal sustainability, it is important to consider the resources of debt repayment, which refer to the government's discretionary financial resources regarding fiscal revenue and the total number of disposable liquid assets. This is expected to eliminate rigid expenditure, which occurs prior to debt repayment. In a word, it is not reasonable to directly judge the local government debt risk according to the primary deficit risk. Secondly, according to a study by Zhang and Zhang [18], the impact of primary deficits has a nonlinear relationship with the local government debt risks in China. Obviously, there is a threshold for different impacts of primary deficits on government debt risks in different intervals, which makes it unsuitable to judge the "sustainability of local government debt" according to "fiscal sustainability" in China.

The first study on the "distance to distress (probability of default)" in the Contingent Claim Analysis (CCA) model was proposed by Gray et al. [19]. Following this, many Western scholars applied the CCA model to the risk analysis of sovereign [20], financial [21], and corporate sectors [22]. Conversely, few scholars have applied it to the field of local governments. For example, a pioneering study by Harris and Piwowar [23] shows that the "distance to distress (probability of default)" is effective for measuring the risks of US municipal bonds. Moreover, an investigation by Silva et al. [24] established a statistical link between the probability of default in each Brazilian state. Since then, following in the footsteps of Harris and Piwowar [23], Chinese scholars have begun to emphasize indigenous study [25–28]. Although scholars apply different details and data compilations when using the CCA model, it is generally believed that the "distance to distress (probability of default)" in the CCA model is an appropriate indicator for debt risks. Because the "distance to distress" can avoid the problem of missing default data in economic entities, it does not need to consider the default losses and bond pricing for local governments. However, it should be noted that Chinese scholars still face several shortcomings in the

application of the "distance to distress", and these shortcomings are mainly reflected in the following three aspects. Firstly, Chinese scholars, like Western scholars, only consider explicit debt and ignore implicit debt in the CCA model. Secondly, some of the "distance to distress" is obtained by a modified CCA model, in which the asset is replaced by fiscal revenue, causing the research to be conducted from a flow perspective, rather than a stock perspective, e.g., Zhou and Hui [25] and Zhuo et al. [26]. Thirdly, different from Harris and Piwowar [23], who used the percentage of average absolute interdealer price concession that is a vacant price index in the Chinese market, and Silva et al. [24], who used the net current revenue volatility from a flow perspective rather than a stock perspective, Liu [27] and Wang et al. [28] used the volatility of treasury bonds to substitute the equity volatility $\sigma_J$ in the CCA model, leading to a series of conclusions not in line with the actual case in China.

With regard to the "risk index system of local government debt", several studies on risk index systems have been carried out in Western countries, such as the "local fiscal monitoring plan and financial crisis method" in America, the "local government borrowing limitation" in Brazil, and the "traffic lights system" in Columbia. In addition, new indexes have been incorporated to make the local government debt risk index system more comprehensive, e.g., Dregger and Kholodilin [29], Wijayanti and Rachmanira [30], and Debra and Falilou [31]. Moreover, the weighting method and construction method of the risk index system have been constantly updated, such as the Bayesian model averaging studied by Kamra [32], event analysis conducted by Balteanu and Erce [33], market pressure approach proposed by Boonman et al. [34], and dynamic signal extraction and dynamic-recursive forecasting technique developed by Dawood et al. [35]. However, there is no sign of the development of an official risk index system in China. Therefore, many Chinese scholars directly copy successful foreign practices in terms of the selection and weighting method of indexes, making their conclusions lack a deep consideration of China's actual case [36]. What is more, the proxy indicators of local government debt risks have been selected from risk index systems, such as the debt burden rate(debt balance/ gross domestic product), debt ratio(debt balance/ government's comprehensive financial resources), or four Basel II indicators which are cash surplus for overheads< 0, legal borrowing limit exceeding 110% of current revenues, solvency< 1 and gross budget savings< 0, to study the local government debt risks, meaning that the research ignores the complexity of local government debt risks.

Since it is inappropriate to take the "scale of local government debt" as a substitute for local government debt risks or use the "sustainability of local government debt", and the index system of local government debt risks in China is a vacant field, the "distance to distress" is considered a good choice. Although there are some shortcomings in setting the equity volatility $\sigma_J$ in the CCA model, the "distance to distress" can be chosen scientifically and objectively as an indicator to analyze the local government debt risks.

### 2.2. Spatial Spillover Effect

Recently, the following has generally been presented by scholars: "The local government debt risks are influenced by the factors such as financial system factors, social and economic factors, political factors, demographic factors and debt management factors" [37–43]. In their studies, it has been found that local governments are regarded as independent risk carriers, while traditional econometric models are applied to explore the impact of macro factors on local government debt risks.

Clearly, China is a vast country with different resource endowments. Therefore, the individual risk of local government debt is important, but the contagion is also vital; otherwise, the study will lack deep thinking on the spatiality of local government debt risks.

Fortunately, scholars have recently begun to consider the geographic distance to analyze the spatial spillover effect of local government debt risks, e.g., Wei [44], Ferraresi et al. [45], and Balaguer-Coll and Ivanova-Toneva [46]. However, their empirical methods have always been

limited to spatial econometric models, such as the spatial autoregressive model, spatial lag model, spatial error model, and spatial Durbin model. Moreover, these models are conclusive regarding studies on the generation mechanism and spatial spillover effect of local government debt risks, but become a liability when it comes to the setting of the model. For example, the statistical inference of spatial autoregressive model is based on the asymptotic property, and normality of the estimated parameter distribution can no longer exist in the small sample [47]. According to the conclusions of Anselin [48], the estimator of spatial lag model is biased and inconsistent based on the least squares method; additionally, an unbiased but invalid estimator has been obtained based on the least square method for spatial error model. Furthermore, an investigation on the small sample property of maximum likelihood(ML), generalized method of moments(GMM), quasi maximum likelihood(QML), and counterpart spatial effect estimation methods has been presented by Zhang [49]. Zhang found that unbiasedness is a very important prerequisite for spatial measurement, but is difficult to achieve by the maximum likelihood estimator, and the asymptotic equivalence of the Moran test, Wald test, likelihood ratio(LR) test, and Lagrange multiplier(LM) test in the small sample may lead to conflicting empirical results. The limitation of the spatial Dubin model(SDM) model has also been reported by Elhorst [50]. The spillover effects are global by construction ($\rho \neq 0$), and global spillovers are often more difficult to justify than local spillovers. If it can be argued theoretically or substantively that "the global spillover effect is more likely than the local one", the SDM model can be taken as a point of departure. Therefore, it is of great significance to adopt another empirical method to study the spatial correlation characteristics and spillover effect of China's local government debt risks.

*2.3. Comparison with Previous Studies*

Another empirical method is applied to analyze the spatial correlation characteristics and spillover effect of China's local government debt risks, which may differ from previous studies in the same field in terms of several points, as follows.

Firstly, there is no escaping the fact that the implicit debt has been ignored in the local government macro balance sheets complied by a few scholars. In terms of the "distance to distress", it is generally believed that the volatility of treasury bonds is seen as the equity volatility in the CCA model, which is the opposite of China's reality. In comparison to traditional studies, a more comprehensive macro balance sheet of local governments is analyzed and the calculation of "distance to distress" is improved.

Secondly, most previous studies on the influencing factors of local government debt risks are limited to traditional macro factors, such as fiscal decentralization, urbanization, and gross domestic product(GDP), which provide a weak explanation on the spatial spillover effect of local government debt risks. As a result, this paper is devoted to clarifying the driving forces behind the spatial centrality of local government debt risks from the other perspectives of local government competition, the division of powers and responsibilities, and local government intervention, with a further analysis of the spatial spillover mechanism of local government debt risks in China.

Thirdly, the available literature focuses on spatial econometric models. Conversely, in this paper, the social network model aims to investigate the spatial spillover effect of local government debt risks in China. Clearly, the provincial local governments in China are similar to the actors in the social network, in line with the definitions given by Zema and Sulich [51]; that is, "each actor in the network has its own legal personality" and "the inter-municipal network is created as a result of willingness to cooperate or compete and the desire for autonomy or relationship". Moreover, the social network model is suitable for risk analysis, crisis management, and political-administrative management [51], and has been successfully applied by Joachim [52], Chen et al. [53], Baird [54], and Zhang et al. [55]. Therefore, it is appropriate to study China's local government debt risks by the social network model.

## 3. Hypotheses

Since the foundation of New China, a new pattern of administrative divisions has been adjusted to satisfy the needs of national construction. With the deepening of reform and opening-up, regulation of the administrative region has reached three levels of divisions referring to provinces (autonomous regions and municipalities directly under the central government), counties (autonomous counties and cities), and townships (ethnic townships and towns). The first level includes 22 provinces, 5 autonomous regions, and 4 munici- palities in mainland China. To be specific, the 22 provinces are Hebei, Shanxi, Liaoning, Jilin, Heilongjiang, Jiangsu, Zhejiang, Anhui, Fujian, Jiangxi, Shandong, Henan, Hubei, Hunan, Guangdong, Hainan, Sichuan, Guizhou, Yunnan, Shananxi, Gansu, and Qinghai, respectively; the 5 autonomous regions are Inner Mongolia, Guangxi, Tibet, Ningxia, and Xinjiang, respectively; the 4 municipalities are Beijing, Shanghai, Tianjin, and Chongqing, respectively; except for the 31 first-level administrative divisions for the mainland, China has another province outside the mainland, namely, Taiwan, and two special administrative regions, namely, Hong Kong and Macao. In total, there are 31 "provincial-level administra- tive regions" in China, which are also called "provinces" by Chinese scholars. It should be noted that Tibet has been deleted from the sample due to its insufficient data disclosure, namely, this paper actually sampled 30 provinces. Furthermore, the local governments are appointed as the leadership by the central authority, and have regional administrative powers related to the unity of national sovereignty under the constitution, while the central government is entitled to exercise the national sovereignty independently and authorize the local government.

In 1994, the reform of China's tax system had a profound impact on China's local governments. Firstly, the reform defined the status of China's local governments, stating that local governments are the main suppliers of local public goods and services, whose fiscal revenue mainly comes from the income obtained for providing public goods and services, such as taxes and fees. Although the autonomy of China's local governments has been expanded to some extent, the increasing tasks of economic construction and public management have resulted in heavy local financial burdens. When fund shortages are booming in local governments, it is inevitable that local governments will be inclined to spend with debt financing. Secondly, the reform has changed the relationships among China's local governments, which has transformed the role of local governments from owners of local state-owned enterprises and township enterprises to tax collectors of lo- cal enterprises. Since then, the competition among China's local governments has been enhanced by setting up administrative barriers to compete for financial resources, repre- sented by tax competition and investment competition. In academia, the former type of competition is named "a lower level of competition within the scope of price competition", and the latter is referred to as "a higher level of competition within the scope of non-price competition, as the main type of competition among Chinese local governments" [56]. Theoretically, these two kinds of competition are supposed to have profound impacts on the generation of local government debt risks and the spatial spillover effect. Specifically, in order to retain tax sources, local governments will take various measures to reduce the actual tax burden of enterprises in their jurisdictions, resulting in a reduction of the tax revenue and fiscal revenue. As a result, local governments will be short of financial resources for providing public goods and services. In order to solve the problem, local governments are committed to issuing local government bonds with transferring land, and local financing platforms tend to issue urban investment bonds. During the whole operation of borrowing business, the behavior of the local government or the function of the local financing platform ignores the solvency of the local governments. Once local gov- ernments benefit from such off-balance-sheet financing, they will continue the borrowing to fake the competitive advantage, with the result of an increase of local government debt risks. Similarly, the impact of investment competition on local government debt risks is mainly represented by attracting foreign direct investment (FDI) [57]. The whole story begins with attracting FDI to cover the large-scale infrastructure construction, forming

a gap between fiscal revenue and expenditure. Then, irrational borrowing starts, which generates a large quantity of local government debt and a heavy burden of debt repayment, ending the story with increased local government debt risks. It should be noted that the competition among local governments will not only increase local government debt risks in their own jurisdictions, but also break geographical boundaries and create multiple network correlations, as well as spatial spillover effects, among local governments. In fact, the competition among local governments is more like a yardstick competition, with such bodies constantly comparing themselves and imitating each other [1]. Taking FDI as an example, a local government's efforts to attract FDI will influence its neighboring local government's investment policy. When a local government sends out a policy signal to attract FDI, the neighboring local governments will start to seek a more rapid overall solution, such as expanding the market and investment scale as a goal, rather than adopting a more efficient and economical measure, in order not to be defeated in the competition. In the end, both the local government and its neighbors increase the scale of local government debt blindly, which is the spatial spillover effect of local government debt risks.

In the face of the 2008 global financial crisis, China has implemented proactive fiscal and monetary policies. Based on government credit, many local governments and local financing platforms have borrowed a lot beyond their solvency. Therefore, the flow of large-scale funds to local governments and local financing platforms has enhanced the debt risks. Meanwhile, in response to the State Council's policy of "insisting on giving priority to transportation infrastructure construction", China's local governments have increased their enthusiasm for investments in infrastructure construction, even running into debt in order to invest. The absence of strict approval and repayment constraints for these debt funds has resulted in rapidly prominent local government debt risks. In theory, under the same macroeconomic environment with the same economic policies of local governments, this is proposed to be a link among China's local government debt risks. Based on the above analysis, the following hypotheses are proposed:

**Hypothesis 1.** *China's local government debt risks have spatial characteristics, such as multiple network correlations.*

**Hypothesis 2.** *China's local government competition has an impact on the spatial centrality and spillover effect of local government debt risks.*

Furthermore, under China's current official selection system, there is the division of powers and responsibilities in local government borrowing, which is vulnerable to the maturity mismatch of local government debts [58]. In terms of the tenure of local government officials in China, provincial and ministerial officials generally serve for 3 to 5 years, while the periods of local government bonds and urban investment bonds last for up to 5 years or more. The inconsistency between the tenure of officials and the period of debts is beneficial for China's local government officials to maximize their own interests. In other words, the local government officials have launched many local projects by continuously accumulating debt, in order to obtain the "achieved projects". Additionally, the "achieved projects" completed during the tenure are regarded as a bargaining chip for political promotion. Once an official of a local government is promoted successfully in this way, officials from the other local governments are expected to copy the shortcut. Thereupon, such a story of blind borrowing to transfer debt repayment responsibilities to the next local government will be repeated among regions, further aggravating the spatial spillover effect of local government debt risks.

Lastly, local government interventions will also intensify local government debt risks and the spatial spillover. In reality, local governments and local financial institutions share many interests intersecting in China. For example, the local government assumes the role of the controlling shareholder of many urban commercial banks, with the right to intervene in the investment decisions of local financial institutions. The common administrative interventions of local governments include personnel appointments, the removal of local fi-

nancial institutions, and fiscal deposits. As another example, the main underwriters of local government bonds and urban investment bonds are the investment banking departments of commercial banks and security companies, as well as their local branches. The major purchasers are the asset management departments and self-operating departments of commercial banks and security companies, especially those departments of urban commercial banks, who have become the main force in purchasing local government bonds and urban investment bonds [59]. Compared with other bond issuers, local governments and local financing platforms do not disclose information in a comprehensive and timely manner, making it difficult for local financial institutions to conduct effective investigations. In addition, there are few qualified and widely recognized credit rating agencies in the market, which makes it difficult for local financial institutions to form accurate judgments on local governments through credit rating agencies. However, under administrative intervention, local financial institutions are willing to continuously purchase local government bonds or urban investment bonds, even if they are in the position of information asymmetry. Meanwhile, based on the above analysis, a local government will imitate the administrative behaviors of neighboring local governments in the race to win. When a local government releases a signal to intervene in purchases of local government bonds or urban investment bonds from financial institutions, no matter if it is true or not, neighboring local governments are urged to strengthen similar administrative interventions, further intensifying the spatial spillover effect of local government debt risks. Based on the above analysis, the following hypotheses are proposed:

**Hypothesis 3.** *The division of powers and responsibilities has an impact on the spatial centrality and spillover effect of local government debt risks in China.*

**Hypothesis 4.** *Local government intervention has an impact on the spatial centrality and spillover effect of local government debt risks in China.*

## 4. Measurement of Local Government Debt Risks based on a Macro Balance Sheet

Debt risks have become a significant challenge in economic growth, which stems from the contradiction between the uncertainty of asset income and the sustainability of debt repayment. Comprehensively, the local government debt risks are not only closely linked to the liabilities of local governments, but also the assets of local governments. Therefore, it is necessary to compile a macro balance sheet of local governments to measure the local government debt risks of various provinces in China.

### 4.1. Liabilities

This paper refers to the compilation method of a national balance sheet presented by Li Yang et al. [60], and the statistical method of local government debt reported by Mao and Huang [61]. Specifically, in this paper, the liabilities of local governments are composed of local government bonds, debt re-loans, and urban investment bonds, as shown in Table 1.

**Table 1.** Local government's macro balance sheet.

| Assets | Liabilities |
|---|---|
| Local state-owned resource assets | Local government bonds |
| Local fiscal deposits | Debt re-loans |
| Local infrastructure assets | Urban investment bonds |
| | **Equity** |
| | Assets—Liabilities |

The original data on the local government bonds were derived from the Wind database following the query path of "Wind database _Risk control topic _Bond risk control _Local government debt _Local government bond statistics". In order to avoid information lapses,

the collected data were collated by the Stata.14 software, with a comparison of the data from the Hexun website. Finally, the sum of the outstanding local government bonds of one province at the end of year $t$ was calculated as the balance of the local government bonds of the province in the $t$ year. Owing to spatial constraints, the detailed information is presented in Appendix A.

Meanwhile, the balance of debt re-loans of every province in year $t$ refers to the sum of its outstanding debt re-loans, for which data are available from the "Finance Yearbook of China". For more information, see Appendix B.

As for data on the urban investment bonds, the work of the collection and collation is similar to that of the local government bonds. The raw data were collected by following the query path of "Wind database _ Wind local government debt _ Urban investment bonds statistics _ Stock bond details", and collated by the Stata.14 software, which tended to be more complete after being compared to the East-Money Choice terminal data. The sum of the outstanding urban investment bonds of one province at the end of year $t$ presented a complete record of the balance of the urban investment bonds of the province for $t$ year. Please see Appendix C for more details.

*4.2. Assets*

In reality, a local government faced with a mountain of debt is impossible to be dealt with through bankruptcy and liquidation like a company. Therefore, the assets of the local government's macro balance sheet in this paper only contain the assets which can be used to protect against debt risks. According to a previous study by Zhao and Yang [62], the assets of local governments are composed of the local state-owned resource assets, local fiscal deposits, and local infrastructure assets, as shown in Table 1.

In fact, all of the natural resources defined as local state-owned resource assets are controlled and operated by the local government, including land, minerals, forests, water, grasslands, and oceans, which cannot be traded or sold directly. According to Chinese laws, transferring the use right of these local state-owned resources is the only way to produce profit for the local governments. Therefore, "land finance" has become extraordinarily popular among the natural resources for the local government. Therefore, in terms of the local state-owned resource assets, this paper only concentrates on the value of land reserved by local governments, which is equal to the area of reserved land multiplied by its unit price [63], as shown in Equation (1):

$$LSRA_{it} = LTTP_{it} \times 3, \tag{1}$$

where $LSRA_{it}$ refers to the local state-owned resource assets of province $i$ in the $t$ year, and $LTTP_{it}$ is subject to the land transfer transaction price of province $i$ in the $t$ year. The multiplier is set to 3 because the duration of the reserved land is between one and three years. Data on the land transaction price come from the Qianzhan database.

Referring to the practice of Zhao and Yang [62], the local fiscal deposits of each province can be estimated by Equation (2):

$$LFD_{it} = \frac{LFR_{it}}{NR_t} \times TLFD_t, \tag{2}$$

where $LFD_{it}$ means the local fiscal deposits of province $i$ in the $t$ year, $LFR_{it}$ serves as the local fiscal revenue of province $i$ in the $t$ year, $NR_t$ is China's national revenue in the $t$ year, and $TLFD_t$ is China's total local fiscal deposits in the $t$ year. The data in Equation (2) are from the Wind database.

According to the method proposed by Jin [64], the local infrastructure assets data can be estimated by Equation (3):

$$K_{i,t+1} = K_{i,t}(1 - \delta) + I_{i,t}, \tag{3}$$

where $K_{i,t+1}$ and $K_{i,t}$ represent the infrastructure capital stock of province $i$ in the $t+1$ year and $t$ year, respectively; $\delta$ plays a role as the depreciation rate; and $I_{i,t}$ describes the fixed asset investment in the infrastructure of province $i$ in the $t$ year. Due to the limited space, the detailed operation is presented in Appendix D.

*4.3. Distance to Distress*

The Contingent Claim Analysis (CCA) was first proposed by Gray et al. [19]. According to Gray et al. [19], the equity is regarded as a call option on assets, while the risky debt is supported as the default-free value of debt minus a put option on assets. Then, the value of equity is computed by the Black–Scholes–Merton formula and Ito's lemma, as shown in Equation (4):

$$\begin{cases} J = AN(d_1) - Be^{-rT}N(d_2) \\ d_1 = \frac{\ln\left(\frac{A}{B}\right) + \left(r + \frac{1}{2}\sigma_A^2\right)T}{\sigma_A\sqrt{T}} \\ J\sigma_J = A\sigma_A N(d_1) \\ d_2 = d_1 - \sigma_A\sqrt{T} \end{cases}, \tag{4}$$

Where $J$ represents the equity market value, $\sigma_J$ refers to the equity volatility, $B$ is defined as the distress barrier, and $r$ describes the risk-free interest rate. Additionally, $A$ illustrates the value of implied assets and $\sigma_A$ means the implied asset volatility.

Mathematically, by inserting the equity market value $J$, equity volatility $\sigma_J$ distress barrier $B$, and risk-free interest rate $r$ into Equation (4), the implied value of two unknowns (implied assets $A$ and implied asset volatility $\sigma_A$) can be computed by iteration. Additionally, the $d_2$ in Equation (4) denotes the "distance to distress" $DD$, and the closer the distance to distress is, the bigger the risk will be, and vice versa.

In practice, Chinese scholars usually set the distress barrier $B$ as Equation (5). Here, $STD$ is the short-term debt which needs to be repaid within one year and $MLTD$ represents the medium and long-term debt whose repayment term exceeds one year.

$$B = STD + \frac{1}{2}MLTD, \tag{5}$$

In addition, Chinese scholars always treat the equity of local governments as a kind of bond, and estimate the equity market value of province $i$'s local government by Equation (6) [27,28]:

$$J_{mi}(t) = \frac{\dot{j}_{mi}(t)}{\dot{j}_{bi}(t)} \times J_{bi}(t), \tag{6}$$

Where $J_{mi}(t)$ serves as the equity market value of province $i$'s local government in the $t$ year, and $J_{bi}(t)$ shows the equity book value of province $i$'s local government in the $t$ year. Meanwhile, $\dot{j}_{mi}(t)$ refers to the market value of local government bonds of province $i$ in the $t$ year, while $\dot{j}_{bi}(t)$ describes the book value of local government bonds of province $i$ in the $t$ year. Moreover, $\dot{j}_{mi}(t)$ is calculated by multiplying the annual closing price of local government bonds by its corresponding issuance volume, and the total issuance volume of local government bonds is combined as $\dot{j}_{bi}(t)$. All relevant data are obtained from the Wind database.

Obviously, it bothers Chinese scholars to set the equity volatility $\sigma_J$. Therefore, the volatility of treasury bonds serves as an alternative in the CCA model, e.g., Liu [27] and Wang et al. [28]. Surprisingly, it has been found that the estimators of $DD$ of China's various provinces are too similar when using this method. Why does the result not conform to the actual case in China? It has been found that $\sigma_J$ has a great impact on $d_2$ in Equation (4). Since the panel data of this paper come from 30 provinces in China from 2009 to 2018, and the $\sigma_J$ values of all provinces are the same, it is easy to develop similar estimators of $DD$ among all provinces. Therefore, it is necessary to improve the setting of $\sigma_J$ to make the $DD$ more applicable and objective, especially for panel data.

Fortunately, on June 3 2020, the Shenzhen Stock Exchange published a series of regional local government bond indexes of China's 15 provinces, which are Zhejiang, Jiangsu, Guangdong, Sichuan, Shandong, Guizhou, Henan, Hebei, Hunan, Hubei, Yunnan, Fujian, Liaoning, Anhui, and Shanxi, respectively. It is easy to obtain the local government bond indexes for 15 provinces. According to the compilation method which can be obtained by following the query path of "CNI index_ Index Series_ Local Government Bonds of Jiangsu Province (921112)_ Information Download", the local government bond indexes of 30 provinces in China can be obtained by Equation (7):

$$GI_t = GI_{t-1} \times \frac{\sum_i (P_{i,t} + AI_{i,t})Q_{i,t-1}}{\sum_i (P_{i,t-1} + AI_{i,t-1})Q_{i,t-1}}, \tag{7}$$

where $GI_t$ represents an explanation of the point of the full price index on $t$ day, and $GI_{t-1}$ means the closing point on $t-1$ day; $P_{i,t}$ and $P_{i,t-1}$ indicate the clean price of the sample bond on $t$ and $t-1$ day, respectively; $A_{i,t}$ and $A_{i,t-1}$ show the accrued interest of the sample bond on $t$ and $t-1$ day, respectively; the amount of the sample bond in circulation on $t-1$ day is labeled as $QI_{i,t-1}$; and the index base day is the first trading day in 2009 with a basis point of 100. Above all, the quotation in the inter-bank bond market is chosen as the price of the sample bond; if no quotation is given in the inter-bank bond market, the valuation price from the CNI index will be used as a substitute. The index of volatility for 30 provinces is easy to obtain by calculating the local government bond indexes. Moreover, with reference to the practice of Liu [27] and Wang et al. [28], these different volatilities in different provinces can be differentially substituted for $\sigma_J$.

Based on the equations of (4)–(7), the "distance to distress" $DD$ can be estimated and improved by setting the equity volatility $\sigma_J$. Then, a social network model can be applied to identify the spatial correlation characteristics and spillover effect of local government debt risks in China.

## 5. Empirical Results

The method of social network analysis is known as a hot research area involving graph theory and mathematical models. In this paper, the social network of local government debt risks describes the relationship of debt risks among local governments, while the spatial network of local governments is composed of nodes and lines of a social network in which the nodes represent each province and the lines represent relationships. Subsequently, this paper captures the relationships and characteristics in the social network based on the gravity model, in order to analyze the spatial centrality of local government debt risks and the spillover effect in terms of the aspects of local government competition, the division of powers and responsibilities, and local government intervention.

### 5.1. Gravity Value

The gravity model has been widely used to study the spatial interaction. According to Newton's law of universal gravitation, gravity proportional to the mass of each object and inversely proportional to the distance between objects. Two basic elements of the model, i.e., the attraction of the region and the distance between regions, are necessary in the application of the gravity model to the field of spatial interaction [65]. In addition, every region has a certain attraction, covering many meanings, such as the economy [66,67], population [68], knowledge production [69], and urban planning [70,71]. Correspondingly, the meaning of distance can not only be the actual distance, but also the integration of time, cost, and other factors.

The basic form of the gravity model is shown in Equation (8):

$$R = \frac{GXY}{d^2}, \tag{8}$$

where $R$ means the interaction between the two masses $X$ and $Y$, and $d$ stands for the distance between two objects. $G$ is denoted as the universal gravity coefficient. Naturally,

it is determined to prove an interaction between objects in the same way as regions. Many scholars have modified Equation (8) to make it applicable to regions. The modified model is shown in Equation (9):

$$R_{ij} = G \frac{X_i^{\beta_1} Y_j^{\beta_2}}{DIS_{ij}^2}, \tag{9}$$

where $R_{ij}$ describes the quantity of spatial flows, $X_i$ and $Y_j$ respectively represent the GDP of two regions, and $DIS$ stands for the distance between two regions. $G$ is denoted as the gravity coefficient, and $\beta_1$ and $\beta_2$ are constants. To complete the picture of a region more scientifically, scholars have taken the population and area of a region into consideration when modifying the gravity model [72]. Moreover, different from objects in physics, the economic impact of region $i$ on region $j$ is not the same as the economic impact of region $j$ on region $i$. The contribution of region $i$ is completely different from that of region $j$ in terms of the economic gravity. Therefore, it is necessary to modify the gravity coefficient $G$, as shown in Equation (10):

$$R_{ij} = k_{ij} \frac{\sqrt{PEO_i GDP_i} \sqrt{PEO_j GDP_j}}{DIS_{ij}^2}, \ k_{ij} = \frac{GDP_i}{GDP_i + GDP_j}, \tag{10}$$

where, $PEO_i$ and $PEO_j$ describe the population in region $i$ and region $j$, respectively; $GDP_i$ and $GDP_j$ mean the GDP in region $i$ and region $j$, respectively; $k_{ij}$ is the modified gravity coefficient, representing the ratio of GDP in region $i$ to the GDP in region $i$ and region $j$; and $R_{ij}$ and $DIS$ have the same meaning as in Equation (9).

Afterwards, more and more scholars began to replace the variables in the application of the gravity model. For example, a study by Lin [66] identifies the spatial structure of tourism economy connections among nine prefecture-level cities of Fujian by replacing *GDP* with total tourism income and replacing *PEO* with total tourist reception. Another investigation on the relationship between foreign portfolio investment (FPI) and economic growth has been conducted by Mazur et al. [67], in which *PEO* is replaced by FPI. A study reported by Li et al. [73] replaces *GDP* and *PEO* with the centrality of nodes to identify influencers in complex networks. Therefore, in order to introduce the gravity model into the study of local government debt risks, this paper can produce a modified gravity model, as shown in Equation (11):

$$R_{ij} = k_{ij} \frac{\sqrt[3]{\frac{1}{DD_i} PEO_i PGDP_i} \sqrt[3]{\frac{1}{DD_j} PEO_j PGDP_j}}{DIS_{ij}^2},$$
$$k_{ij} = \frac{\frac{1}{DD_i}}{\frac{1}{DD_i} + \frac{1}{DD_j}} = \frac{DD_i \times DD_j}{DD_i (DD_i + DD_j)} \tag{11}$$

where $R_{ij}$ represents the gravity value of two provinces, with the gravity coefficient of $k_{ij}$, and $DIS_{ij}$ measures the distance between province $i$ and province $j$, which refers to the shortest distance between the provincial local governments on the Baidu map. *DD* is given as the "distance to distress" which is taken as the reciprocal in Equation (11) due to its negative indication, and *PEO* counts the resident population from the Wind database, whereas *PGDP* represents the GDP per capita from the Wind database.

According to Equation (11), spatial correlations of local government debt risks in China's 30 provinces can be represented by the gravity value, and then expressed by a matrix. Afterwards, this matrix can be binarized based on the average of elements in the matrix since the first 5% and last 5% of the data have been deleted to reduce the influence of extreme values. Specifically, all of the elements in the matrix larger than the average will be set to 1, indicating that there is a local government debt risk correlation between $i$ and $j$, whereas all of the elements less than the average will be set to 0, indicating that there is no local government debt risk correlation between $i$ and $j$. After binarizing the spatial correlation matrix, the spatial correlations of China's local government debt risks

from 2009 to 2018 could be obtained, as shown in Figure 1a,b (data were calculated by the binarized matrix mentioned above with the help of Ucinet 6.560, and drawn by the ArcMap 10.5). Due to the limited space, only two figures for the year 2009 and 2018 are shown, while the rest are presented in Appendix E.

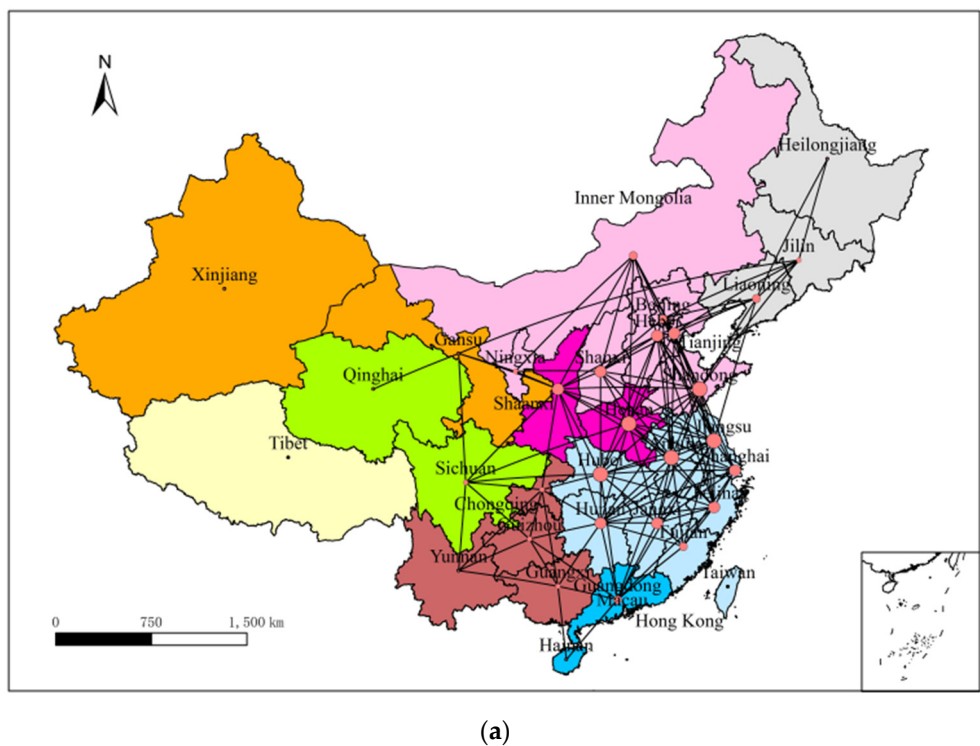

(**a**)

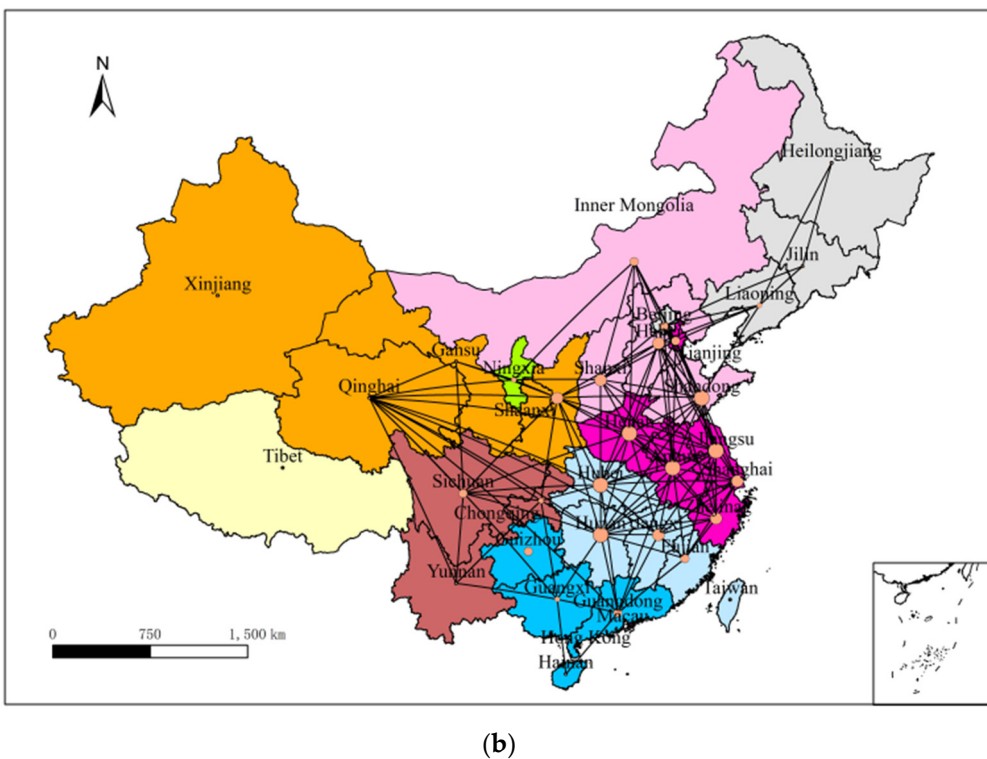

(**b**)

**Figure 1.** (**a**) Spatial characteristics of China's local government debt risks in 2009. (**b**) Spatial characteristics of China's local government debt risks in 2018.

Except for a few provinces, such as Xinjiang, which was isolated for several years, there are complex and multiple spatial correlations between local government debt risks of China's 30 provinces in the figures, which makes it difficult to cut off the contagion of debt risks without all-round monitoring for all provinces. Consequently, Hypothesis 1 is verified. In addition, the spatial correlations show some heterogeneity. It is shown that Jiangsu, Hunan, Shaanxi, Tianjin, and other provinces have more spatial correlations, while Xinjiang, Qinghai, Hainan, and other provinces have fewer spatial correlations. Furthermore, by comparison, it has been found that provinces with more spatial correlations have a larger economic volume to exert a greater impact on China's macroeconomic stability. Therefore, once a serious local government debt risk occurs in such a province, it will quickly spread to other provinces through the spatial correlations and lead to a wide range of local government debt crises in China.

### 5.2. Spatial Correlation Characteristics

In the previous section, the spatial correlations between local government debt risks were described by the lines in the figures. Several analyses are presented in the following section, including the eigenvector centrality analysis, subgroup analysis, and connectedness and small world analysis, in order to better capture the spatial characteristics of China's local government debt risks.

### 5.2.1. Eigenvector Centrality Analysis

The eigenvector centrality is shown in Equation (12) and described by the area of the dot in the figures:

$$EC_i = x_i = \lambda^{-1} \sum_{j \in M(i)} x_j = \lambda^{-1} \sum_{j \in N(g)} a_{i,j} x_j, \tag{12}$$

Where $N(g)$ refers to the given network and $g$ means the sum of nodes. Meanwhile, $A = a_{i,j}$ is the adjacency matrix of the network $N(g)$. To be more specific, $a_{i,j} = 1$ if node $i$ is linked to node $j$, and $a_{i,j} = 0$ otherwise. $M(i)$ is studied by setting the neighbors of node $i$, while $\lambda$ presents the largest eigenvalue of $A = a_{i,j}$ according to the Perron–Frobenius theorem. It is noted that the larger $EC_i$ is, the more central node $i$ will be, and the larger the area of the dot will be in the figures.

From the figures, it is found that the central nodes run from China's eastern coastal provinces to the central provinces, especially the eastern coastal provinces, while the western provinces remain slim. The eigenvector centrality of each province from 2009 to 2018 is ranked as follows: (i) For Hunan, a province in the central region, its rank has risen significantly since 2015, and jumped to fourth place in 2018; (ii) in 2013, a province from the western region broke into the top ten for the first time, namely, Shaanxi, and this province has remained at the top since then; (iii) in western regions, the most noteworthy provinces are Guizhou and Yunnan. To be specific, the rank of Guizhou rose sharply in 2017 and continued until 2018. The reason for this is that Guizhou is in a critical period of poverty alleviation. Therefore, Guizhou relies on the government's borrowing to maintain its huge investment in high-speed rail, highways, and big data, causing a significantly higher risk than that of neighboring provinces. A similar situation happened in Yunnan, whose rank rose sharply in 2017 due to historical reasons and limitations of its own economic development.

### 5.2.2. Subgroup Analysis

The above eigenvector centrality analysis is beneficial for identifying the central nodes and trends of China's local government debt risks. However, how do provincial local government debt risks with different centralities cluster together? What is the relationship between subgroups? With the help of the convergent correlation (Concor) in Ucinet6.560, combined with a maximum partition of 3, a convergence criterion of 0.2, and a maximum iteration of 25, the results shown in the figures demonstrate that the provinces with the same color are in the same subgroup.

The eight subgroups are similar to the seven major geographic regions of China, which consist of Northeast China (Heilongjiang, Jilin, and Liaoning), East China (Shanghai, Jiangsu, Zhejiang, Anhui, Fujian, Jiangxi, and Shandong), North China (Beijing, Tianjin, Shanxi, Hebei, and Inner Mongolia), Central China (Henan, Hubei, and Hunan), South China (Guangdong, Guangxi, and Hainan), Southwest China (Chongqing, Sichuan, Guizhou, Yunnan, and Tibet) and Northwest China (Shaanxi, Gansu, Qinghai, Ningxia, and Xinjiang), but not exactly the same. Specifically, the three northeast provinces are in the same subgroup, with a density of 1. North China belongs to the same subgroup, with a density of 0.855. East China is divided into two parts: The north part is a subgroup, and the south part combined with Central China forms a whole subgroup. Additionally, the structure of the north part is not stable, while the structure of the south part is relatively stable, with a density of 0.984. Southern China belongs to the same subgroup, with a density of 0.826. The southwest region is in the same subgroup, with a density of 0.83. The northwest region is mainly comprised of two cores: One is Xinjiang which is more like an isolated node in some years, with a density of less than 0.5, and the other is Qinghai, with a density of only 0.7. Clearly, since 2017, a new trend has appeared in Northwest China, where a unified subgroup has been formed in Xinjiang and Qinghai, but an isolated one has developed in Ningxia. The reason for this is that the debt growth rate of Ningxia has ranked top repeatedly and far exceeded those of neighboring provinces during the "Twelfth Five-Year Plan" period.

Furthermore, in order to study the partition degree of the subgroups, the External-Internal Index ($EI$) is given in Equation (13):

$$EI = \frac{EL - IL}{EL + IL}, \tag{13}$$

Where $EL$ is short for the number of external links, and $IL$ is labeled as the number of internal links. As the $EI$ approaches 1, all of the links will be external to subgroups. A score of $-1$ indicates that all of the links are internal. If the links are divided equally, which means that the links among subgroups are randomly distributed, the $EI$ index will equal 0. The $EI$ index in this paper is shown in Table 2. It should be noted that the significance test for the parameters in Table 2 is shown in Appendix F.

**Table 2.** Spatial correlation characteristics of local government debt risks.

|      | EI | IL | EL | Connectedness | Least Upper Boundedness | Characteristic Path Length | Clustering Coefficient | Transitivity |
|------|------|------|------|------|------|------|------|------|
| 2009 | 0.152 | 0.424 | 0.576 | 0.9333 | 1 | 2.032 | 0.684 | 39.11% |
| 2010 | 0.227 | 0.386 | 0.614 | 0.9333 | 1 | 2.012 | 0.686 | 38.57% |
| 2011 | 0.259 | 0.37 | 0.63 | 0.9333 | 1 | 1.963 | 0.702 | 37.96% |
| 2012 | 0.203 | 0.398 | 0.602 | 0.9333 | 1 | 1.99 | 0.716 | 40.31% |
| 2013 | 0.191 | 0.405 | 0.595 | 0.9333 | 1 | 1.995 | 0.73 | 41.25% |
| 2014 | 0.308 | 0.346 | 0.654 | 0.9333 | 1 | 1.995 | 0.717 | 40.32% |
| 2015 | 0.248 | 0.376 | 0.624 | 0.9333 | 1 | 1.97 | 0.722 | 38.86% |
| 2016 | 0.164 | 0.418 | 0.582 | 0.9333 | 1 | 1.985 | 0.708 | 39.05% |
| 2017 | 0.277 | 0.362 | 0.638 | 0.9333 | 1 | 1.983 | 0.725 | 37.09% |
| 2018 | 0.354 | 0.323 | 0.677 | 0.9333 | 1 | 2.017 | 0.7 | 37.45% |

It can be seen from Table 2 that the $EI$ index shows a volatile increase in the range of 0.152 to 0.354 during the sample period, reflecting a random distribution among subgroups. Comparing the results of internal E-I and external E-I in Table 2, it is found that the external links are greater in number and exhibit a clear upward trend, and tend to increasingly dominate the internal links. From the above analysis of the $EI$ indexes, it can be inferred that if a local government debt risk occurs in a certain subgroup, it will spread from one subgroup to others.

### 5.2.3. Connectedness and Small World Analysis

Theoretically, connectedness and small world analysis is composed of the connectedness (*C*) of Equation (14), the least upper boundedness (*LUB*) of Equation (15), the characteristic path length (*L*) of Equation (16), the clustering coefficient (*CC*) of Equation (17), and the transitivity (*T*) of Equation (18):

$$C = 1 - \left[ \frac{V}{N(N-1)/2} \right], \tag{14}$$

$$LUB = 1 - \frac{U}{\max(U)}, \tag{15}$$

$$L \approx \frac{N}{2K+1}, \tag{16}$$

$$CC \approx 1 - \frac{6}{K^2 - 1}, \tag{17}$$

$$T = \frac{3R}{P}, \tag{18}$$

where *V* means the number of unreachable pairs in the network, and *N* covers the size of the network. Additionally, *U* illustrates the number of pairs that have no *LUB*, and *K* represents the mean degree of nodes, with the number of triangles of *R* and the number of all triplets of *P*.

As shown in Table 2, (i) the connectedness during the sample period is stable at 0.9333, again verifying the strong spatial correlations of China's local government debt risks; (ii) the least upper boundedness always operates on the result of 1, reflecting that the graph hierarchy is very high, which is consistent with the central province effect mentioned above; (iii) the characteristic path length exhibited a maximum of 2.032 in 2009, and a minimum of 1.963 in 2011, both of which are less than the critical value of 10; (iv) the maximum value of the clustering coefficient is 0.73 in 2013, and the minimum value is 0.684 in 2009, both of which are beyond the critical value of 0.65. Overall, China's local government debt risks display a "small world" architecture with a distinctive combination of a large clustering coefficient and short characteristic path length, which is evident from 2011 to 2017. This also reflects that the spatial correlations of China's local government debt risks have a compact structure during 2011–2017; and (v) after reaching its peak in 2013, the transitivity presents a volatile decline, accompanied by a drop of local government risks from 2014, which is in line with China's frequent introduction of local government debt management policies and active resolutions of local government debt risks in recent years.

### 5.3. Spatial Centrality

Different from conventional statistical analysis, Quadratic Assignment Procedure Regression (QAP) is applied to study the spatial centrality of China's local government debt risks influenced by local government competition, the division of powers and responsibilities, and local government intervention. In conventional statistical analysis such as multiple regression analysis, it is pointed out that independent variables as the necessary prerequisite are required to be relatively independent, without being highly linearly correlated; otherwise, multicollinearity will occur, along with a whole suite of other problems. Unlike conventional statistical analysis, QAP does not rely on such a prerequisite; instead, it performs the permutation test to examine the "relationships", rather than describe attributes of the "variables" [74]. To be specific, QAP regression regresses a dependent matrix on one or more independent matrices to assess the significance of the R-square, as well as the regression coefficients. The algorithm proceeds in two steps: (i) It performs a standard multiple regression in response to the dependent and independent matrices, and (ii) the regression is calculated by a random arrangement of rows and columns of the dependent matrix to compute the values of the R-square and all coefficients, which have to be repeated hundreds of times to estimate standard errors. For each coefficient, the program counts the proportion of random permutations that yielded a coefficient as extreme as the one

computed in step 1. In brief, QAP constructs a reference distribution of random parameters derived from a dataset with the same structure as the dataset under evaluation.

The dependent "variable" is defined by the centrality of each province, and the independent "variables" include local government competition, the division of powers and responsibilities, and local government intervention. It is worth noting that the "variables" here are not exactly the variables in conventional statistical analysis, but the "relationships" in social network analysis. The word "variable" is used here to allow readers to understand the empirical research of this paper. The quotation marks encapsulating the word will be removed in the following section.

These variables are described as follows.

The dependent variable is defined by the eigenvector centrality of each province obtained from Equation (12). It should be noted that the larger the eigenvector centrality is, the more spillovers between a local government and other local governments there will be, accompanied by a more obvious central position of the local government.

According to the previous analysis, local government competition is concentrated on attracting FDI. The more competitive a local government is, the easier it is to attract more FDI. At present, various methods are used to construct the indicator of local government competition with FDI. The common indicators include the number of registered FDI enterprises, the FDI registered capital, and the FDI amount. Since the FDI is very vulnerable to the external economic environment, this paper measures the degree of local government competition by the proportion of the FDI of each local government in the total FDI of China, and replaces this proportion by the amount of FDI per capita in the robustness test [75].

According to Article 98 and 106 of the Constitution of the People's Republic of China, it is easy to divide powers and responsibilities of the local government under the current official tenure system in China [76]. Therefore, the larger the proportion of long-term local government debt in total debt, the higher the degree of the division of powers and responsibilities, accompanied by less awareness of the debt risk management of the local government. According to the definition given by Shen et al. [77], long-term debt refers to debt whose payback period is longer than 5 years, and the proportion of long-term local government debt in total debt is chosen as an indicator of the division of powers and responsibilities. In addition, based on the survey of 179 provincial governors in China's 28 provinces conducted by Duan [78], the most common tenure of provincial governors is 3 years. Therefore, local government debt for 3 years or more is introduced into the robustness test.

Theoretically, the stronger the administrative intervention power of the local government, the easier it is to realize the soft constraints of local governments on local financial institutions. Since the local fiscal expenditure is an appropriate indicator for measuring the degree of local government intervention, this paper refers to the practice of Xie et al. [79], and measures the degree of local government intervention by the proportion of financial supervision and other affairs expenditure of each local government in those of the whole nation. In addition, referring to the practice of Shen et al. [77], this paper takes the average value of national local public financial expenditure as a benchmark in the robustness test.

It should be noted that the above dependent variables and independent variables are column vectors, which need to be converted into N × N matrices using the 2-Mode to 1-Mode algorithm. Moreover, this paper adopts the Double Dekker Semi-Partialling MRQAP in Ucinet6.560. The empirical results are shown in Table 3.

It can be seen from Table 3 that both the R-square and the adjusted R-square are 0.266, with the *p*-value of 0. The results show that the model fits a higher degree with a relatively larger R-square and adjusted R-square, as well as an extremely small *p*-value.

**Table 3.** Quadratic Assignment Procedure (QAP) multiple regression.

| | Model Fit | | | | |
| --- | --- | --- | --- | --- | --- |
| | R-Square | Adj R-Sqr | *p*-Value | Random Seed | Perms |
| Model | 0.266 | 0.266 | 0.000 | 944 | 2000 |
| | Regression Coefficients | | | | |
| | Un-Stdized | Stdized Coef | *p*-Value | As Large | As Small |
| Local Government Competition | 5.87344 | 0.46593 | 0.00050 | 0.00050 | 1.00000 |
| Division of Powers and Responsibilities | 0.05314 | 0.11620 | 0.00450 | 0.00450 | 0.99600 |
| Local Government Intervention | 0.07549 | 0.01101 | 0.33583 | 0.33583 | 0.66467 |
| Intercept | 0.01112 | 0.00000 | 0.00000 | 0.00000 | 0.00000 |

It can be seen from Table 3 that both the unstandardized coefficient and standardized coefficient are positive, indicating that local government competition, the division of powers and responsibilities, and local government intervention all have positive impacts on the spatial centrality of China's local government debt risks. Regarding the *p*-values, it can be found that both local government competition and the division of powers and responsibilities have passed the 1% significance test, while local government intervention is not statistically significant. The above results confirm the first part of Hypothesis 2 and Hypothesis 3; that is, the higher the degree of local government competition or division of powers and responsibilities in a certain province is, the greater the possibility of the province being in the spatial central position of local government debt risks will be. In addition, the coefficient of local government intervention is positive, but not statistically significant, indicating that there is a non-significant trend in the first part of Hypothesis 4. Specifically, China's commercial banks have unbreakable faith in the government, especially local urban commercial banks and rural commercial banks. For these banks, local government bonds and urban investment bonds are still excellent investment targets because of the rigid payment. As a result, local government intervention is not the decisive factor for them to become the main force in the purchase of local government bonds and urban investment bonds. Furthermore, in terms of the impact of independent variables, the standardized coefficient is taken as the criterion. Consequently, it can be found that the impact of local government competition is much stronger than that of the division of power and responsibilities, demonstrating that local government competition is the core driving force of the spatial centrality of China's local government debt risks.

*5.4. Spatial Spillover Effect*

To verify the spatial spillover effect of local government debt risks, the symmetric adjacency matrix partitioned into two groups was included in the randomization test of autocorrelation by the Joint-Count program in Ucinet6.560. The routine is limited to two groups. Based upon counting the entries within and between the groups, a comparison with a randomized model was conducted.

In this paper, the spatial correlation matrix of local government debt risks shown in the figures is set as the dependent variable. Accordingly, the independent variable is presented by the column vector of local government competition, the division of power and responsibilities, and local government intervention after binarization. The results are shown in Table 4.

Table 4 shows the expected and observed counts. Generally speaking, "1–1" presents the counts within group 1, while "1–2" describes the counts between the groups, and "2–2" illustrates the counts within group 2. In theory, the expected values refer to the values randomly distributed within or between the groups, and the "Difference" in Table 4 is defined by subtracting the expected from the observed. If the "Difference" is larger than or equal to 0, *p* > = Diff; otherwise, *p* < = Diff.

**Table 4.** Spatial spillover effect.

| Independent Variable | Random Seed | Perms | Group | Expected | Observed | Difference | *p*-Value | *p* > = Diff | *p* < = Diff |
|---|---|---|---|---|---|---|---|---|---|
| Local Government Competition | 31,221 | 10,000 | 1–1 | 511.842 | 358 | −153.842 | 0.000 | 1.000 | 0.000 |
| | | | 1–2 | 621.916 | 574 | −47.916 | 0.012 | 0.990 | 0.012 |
| | | | 2–2 | 186.242 | 388 | 201.758 | 0.000 | 0.000 | 1.000 |
| Division of Powers and Responsibilities | 10,215 | 10,000 | 1–1 | 320.155 | 280 | −40.155 | 0.028 | 0.975 | 0.028 |
| | | | 1–2 | 662.090 | 639 | −23.090 | 0.110 | 0.900 | 0.110 |
| | | | 2–2 | 337.755 | 401 | 63.245 | 0.004 | 0.004 | 0.996 |
| Local Government Intervention | 23,680 | 10,000 | 1–1 | 658.264 | 549 | −109.264 | 0.000 | 1.000 | 0.000 |
| | | | 1–2 | 549.073 | 623 | 73.927 | 0.000 | 0.000 | 1.000 |
| | | | 2–2 | 112.664 | 148 | 35.336 | 0.007 | 0.007 | 0.994 |

It should be noted that the three independent variables have already been binarized based on their averages. Specifically, all the elements in the matrix larger than the average will be set to 1, whereas all the elements less than the average will be set to 0. Then, provinces with a lower degree of local government competition are coded as group "0", whereas provinces with a higher degree of local government competition are coded as group "1". The same treatment is true for the division of powers and responsibilities, as well as local government intervention. The old group code can automatically be translated into the new one by Ucinet 6.560. To be specific, the old group code "0" is labeled as the new group code "1", while the old group code "1" is changed to the new group code "2". Therefore, the "1–1" in Table 4 presents the counts within group 1, standing for the group with a lower degree of local government competition, division of power and responsibilities, or local government intervention, while "2–2" illustrates the counts within group 2, standing for the group with a higher degree of local government competition, division of power and responsibilities, or local government intervention. Moreover, the "Difference" between the observed and expected values of "1–1" and "2–2" can further reflect the impacts of independent variables on dependent variables. Taking the division of powers and responsibilities as an example, if the "Difference" of "1–1" is smaller than that of "2–2", it means that the higher the degree of division of power and responsibilities is, the stronger the spatial spillover effect on China's local government debt risks.

From Table 4, in terms of local government competition, it can be seen that both "1–1" and "2–2" passed the 1% significance test, and "1–2" passed the 5% significant test. The significant test of "1–2" indicates that the spatial spillover effect of local government competition is successfully carried out for China's local government debt risks across groups. In addition, the observed counts are smaller than the expected counts in "1–1", while the observed counts are higher than the expected counts in "2–2", indicating that the higher the degree of local government competition is, the stronger the spatial spillover effect on China's local government debt risks, which verifies the latter part of Hypothesis 2.

In terms of the division of powers and responsibilities, "1–1" and "2–2" passed the 5% and 1% significance test, respectively, and "1–2" passed the 15% significant test. The significance test of "1–2" demonstrates that the spatial spillover effect of division of power and responsibilities on local government debt risks prevails in China. Similarly, from the perspective of expected and observed values, an increase of the division of power and responsibilities will aggravate the spatial spillovers of local government debt risks, which verifies the latter part of Hypothesis 3. More importantly, the *p*-value of "1–2" is relatively large, and a week homophily can be found in the spatial spillover effect within the groups.

In terms of local government intervention, "1–1", "2–2", and "1–2" all passed the 1% significance test. Although local government intervention has no significant positive impact on the spatial centrality of China's local government debt risks, it will have a positive spatial spillover effect on local government debt risks, which verifies the latter part of Hypothesis 4. Additionally, considering the difference between the expected and observed values, it is found that the positive impact is more obvious outside the groups, which is

probably due to the mutual imitation of the administrative decision-making behavior of local governments.

## 6. Robust Test

According to the analysis of the robust test, a new construction was established to test the robustness of the three independent variables, including local government competition, the division of powers and responsibilities, and local government intervention. Then, the QAP multiple regression and spatial spillover effect test were conducted repeatedly. The results are shown in Tables 5 and 6.

**Table 5.** Robust test of QAP multiple regression.

| | **Model Fit** | | | | |
| | **R-Square** | **Adj R-Sqr** | ***p*-Value** | **Random Seed** | **Perms** |
|---|---|---|---|---|---|
| Model | 0.335 | 0.335 | 0.000 | 506 | 2000 |
| | **Regression Coefficients** | | | | |
| | **Un-Stdized** | **Stdized Coef** | ***p*-Value** | **As Large** | **As Small** |
| Local Government Competition | 5.58391 | 0.16984 | 0.00050 | 0.00050 | 1.00000 |
| Division of Powers and Responsibilities | 0.02554 | 0.12791 | 0.00150 | 0.00150 | 0.99900 |
| Local Government Intervention | 0.02978 | 0.47956 | 1.00000 | 1.00000 | 1.00000 |
| Intercept | 0.00645 | 0.00000 | 0.00000 | 0.00000 | 0.00000 |

**Table 6.** Robust test of the spatial spillover effect.

| Independent Variable | Random Seed | Perms | Group | Expected | Observed | Difference | *p*-Value | *p* > = Diff | *p* < = Diff |
|---|---|---|---|---|---|---|---|---|---|
| Local Government Competition | 16,943 | 10,000 | 1–1 | 645.873 | 570 | −75.873 | 0.002 | 0.999 | 0.002 |
| | | | 1–2 | 556.254 | 532 | −24.254 | 0.128 | 0.881 | 0.128 |
| | | | 2–2 | 117.873 | 218 | 100.127 | 0.000 | 0.000 | 1.000 |
| Division of Powers and Responsibilities | 22,994 | 10,000 | 1–1 | 324.511 | 268 | −56.511 | 0.004 | 0.997 | 0.004 |
| | | | 1–2 | 662.178 | 619 | −43.178 | 0.015 | 0.987 | 0.015 |
| | | | 2–2 | 333.311 | 433 | 99.689 | 0.000 | 0.000 | 1.000 |
| Local Government Intervention | 7804 | 10,000 | 1–1 | 427.786 | 240 | −187.786 | 0.000 | 1.000 | 0.000 |
| | | | 1–2 | 649.228 | 607 | −42.228 | 0.018 | 0.984 | 0.018 |
| | | | 2–2 | 242.986 | 473 | 230.014 | 0.000 | 0.000 | 1.000 |

Government competition has decreased, it is still significantly positive and larger than that of the division of powers and responsibilities. The coefficient of the division of powers and responsibilities has risen slightly and is significantly positive. The coefficient of local government intervention is positive, but still fails to pass the significance test. In a word, the behavior of changing the construction method of the independent variables will not affect the robustness of conclusions for the QAP regression.

Comparing Table 6 with Table 4, it has been found that the empirical conclusions from Table 6 are basically consistent with Table 4, in addition to the different significance of local government competition and the division of powers and responsibilities from "1–2". The robust test suggests that local government competition, the division of powers and responsibilities, and local government intervention all have positive impacts on the spatial spillover effect of China's local government debt risks.

## 7. Conclusions and Policy Recommendations

The goal of this paper is to study the spatial correlation characteristics and spillover effect of local government debt risks in China. So far, the methods, such as "sustainability of local government debt", have caused many controversies over the measure. Therefore, this paper has made an effort to detail a compilation of the macro balance sheet of China's local governments, as well as calculate the "distance to distress". With the introduction

of the social network model, evidence pointing towards the existence of multiple and heterogeneous spatial correlations of China's local government debt risks has been found. Concretely, the eastern coastal provinces and central provinces take up the central nodes, while the western provinces exhibit the opposite trend, and more attention should be paid to some central and western provinces, such as Hunan, Shananxi, Guizhou, and Yunnan.

From a regional perspective, the distribution of eight subgroups in this paper is similar to the seven major geographic regions of China, but not exactly the same. The links among subgroups are generally randomly distributed and tend to be external.

In terms of the network structure, China's local government debt risks are featured with characteristics of a "small world", particularly from 2011 to 2017. The finding that the transitivity reached a peak in 2013, before retreating uneasily, indicates that the spread of China's local government debt risks has been slowed down since 2014.

The spatial spillover effect of China's local government debt risks has proved to be influenced by the local government competition, division of powers and responsibilities, and local government intervention. However, the spatial centrality of China's local government debt risks is only influenced by two of the above factors, including the local government competition and division of powers and responsibilities; additionally, the former has a stronger impact than the latter.

Based on the above conclusions, some policy recommendations can be put forward.

First, more emphasis should be put on the multiple spatial correlations of local government debt risks in China, and it is necessary to strengthen the assessment and early warning systems of debt risks, especially for local governments in central and western regions.

Second, in order to alleviate the spillover effect of local government competition on local government debt risks, it is important to abandon the local protectionism and prevent vicious economic competition. Ideally, it will be effective to reduce the behavior of irrational borrowing by establishing a cooperative relationship between local governments.

Third, it is imperative to reduce the spatial spillovers of local governments influenced by the division of powers and responsibilities. For example, it is effective to appropriately extend the tenure of local government officials. Additionally, the focus of the performance appraisal of local government officials should not only be on the achievements during their tenure, but also on the economic and social development of their original jurisdiction within 3 to 5 years after the end of their tenure.

Lastly, the improper intervention of local governments on local financial institutions should be forbidden, as in local financial institutions, where it is a "quasi financial instrument". In addition, China's local governments should develop the conception of "social responsibility" proposed by Rutkowska et al. [80], so as to reduce legal conflicts on complying with regulatory requirements and changing relationships among the local financial institutions. In the meantime, it is imperative to improve the civic awareness of China's residents, and encourage residents to report unreasonable behavior of borrowing, as well as the improper intervention of local governments.

However, limited by the data size, only ten years of data were available for the analysis presented in this paper. Further research is expected to focus on the following points. Firstly, we will keep track of China's data to expand the sample size. Different from previous studies, an early warning system with a combination of the individual risk and contagion risk of local government debt will be designed. Thirdly, the study on the spillover effect of local government debt risks will further explore its vertical reverse relationship with national government debt risks, vertical frontal relationship with risks in the household sector, and horizontal relationship with risks in the financial sector. Lastly, country-oriented research on local government debt risks will be conducted to examine different spatial correlations and the spillover effect of local government debt risks in different countries.

**Author Contributions:** Conceptualization, X.L. and X.G.; methodology, X.L.; software, X.L.; validation, X.L. and W.F.; formal analysis, X.G.; investigation, X.L. and H.Z.; resources, X.L.; data curation, X.L. and H.Z.; writing—original draft preparation, X.L.; writing—review and editing, X.G. and H.Z.;

visualization, W.F.; supervision, X.G.; project administration, X.G.; funding acquisition, X.G. All authors have read and agreed to the published version of the manuscript.

**Funding:** This research is supported by the National Natural Science Foundation of China (Grant No.71901222 and No. 71974204), and the Fundamental Research Funds for the Central Universities, Zhongnan University of Economics and Law (Grant No. 2722020JX005).

**Institutional Review Board Statement:** Not applicable. The study did not involve humans or animals.

**Informed Consent Statement:** Not applicable.

**Data Availability Statement:** Most of the data and programs in this paper can be published on the Internet. Because the collection and collation of local government debt data has taken a lot of time and effort from researchers of financial engineering and risk management research center of Wuhan University (X.L. and W.F. are members of the center), these data will be only provided to review experts and editors for review, and it is not recommended that they are published on the Internet, in order to protect the rights of other scholars in this research center to use the data to publish more papers.

**Conflicts of Interest:** The authors declare that there are no conflict of interest regarding the publication of this paper.

## Appendix A

The following steps should be performed to obtain the local government bonds. Firstly, the raw data on China's local government bonds from 2009 to 2018 from the Wind database should be downloaded by following the query path of "Wind database _Risk control topic _Bond risk control _Local government debt _Local government bond statistics". Secondly, because Wind's raw data have a lot of duplicate exchange information, it is necessary to run a Stata.14 do-file to eliminate this information. Then, since Wind's raw data is a kind of aggregated data, regardless of province, the data of local government bonds in certain provinces need to be assigned according to the bonds' names. Significantly, first, the local government bonds issued on behalf of the Ministry of Finance during 2010~2014 should be merged with provinces according to the bond issuance announcement. Second, the local government bonds issued by Qingdao, Ningbo, and other separately listed cities should be merged with the province to which the city belongs. The reason for this is that although these cities are in the second-level administrative divisions of China rather than the first-level administrative divisions, their economic status is so important that they have been labeled separately by the Ministry of Finance. In order to avoid the omissions in the Wind database, with reference to the solution proposed by Mao and Huang [61], the data should be compared with local government bonds data downloaded from the Hexun website. Finally, the sum of the outstanding local government bonds of a province at the end of year $t$ should be defined as the balance of the local government bonds of the province in the $t$ year.

## Appendix B

This part focuses on debt re-loans data: According to the Notice (No. 479[1999]) of "the Ministry of Finance on Certain Issues Concerning the Repayment of Principals and Interests of Debt Re-loans", the repayment period of debt re-loans in developed coastal areas is 6 years, including a grace period of 2 years, with an annual interest rate of 5.5%, while in the central and western regions, the repayment period is extended to 10 years, including a grace period of 2 years, with an annual interest rate of 5%. Based on the Notice (No. 479[1999]), the data on debt re-loans were collected by the following steps. Firstly, we collected the raw data of debt re-loans from the "Finance Yearbook of China", and incorporate the data of Qingdao, Ningbo and other separately listed cities into the province to which the city belongs. Secondly, we defined the balance of each province's debt re-loans of the $t$ year as the sum of the province's outstanding debt re-loans till the $t$ year. To be specific, the balance of the developed coastal province of the $t$ year was obtained by the amount of debt re-loans from year $t$ to year $t - 5$, while the balance of

the central or western province of the *t* year was calculated from the amount of debt re-loans from year *t* to year $t - 9$.

**Appendix C**

An explanation of urban investment bonds data is shown in Appendix C. Firstly, we collected the raw data of China's urban investment bonds from 2009 to 2018 from the Wind database by following the query path of "Wind database _ Wind local government debt _Urban investment bonds statistics _ Stock bond details". Secondly, because Wind's raw data have a lot of duplicate exchange information, it was necessary to run a Stata.14 do-file to eliminate this information. Fortunately, because Wind's raw data on urban investment bonds have been provided according to province classification, we did not need to conduct a similar step for local government bonds. It is expected that the processed data will be compared with data from the East-Money Choice terminal to fill in the omission. Finally, the sum of the outstanding urban investment bonds of one province at the end of year *t* represents the balance of the urban investment bonds of the province at the *t* year.

**Appendix D**

The local government's infrastructure assets are recorded by the perpetual inventory system in the Jin [64]. According to classification of the "Statistical Yearbook of the Chinese Investment in Fixed Assets", the infrastructure basically refers to economic infrastructure and social infrastructure. Specifically, a suitable economic infrastructure is built with four industries, for instance, "production and supply of electricity, water and gas"; "transport, storage and post"; "information transmission, computer services and software industry"; and "water conservancy, environment and public utilities management". In comparison, the emphasis of improved social infrastructure changes to five industries, such as the "health, social security and social welfare industry"; "education"; "culture, sports and entertainment"; "scientific research, technical service and geological prospecting industry"; and "public administration and social organizations". In order to estimate the local government's infrastructure assets, as shown in Equation (3), we followed the following steps below. Step 1: In order to obtain the $I_{i,t}$, we firstly multiplied the "completion of fixed assets investment by industry" of province *i* in the *t* year by the "utilization rate of fixed assets by industry" to get the "newly-added fixed assets by industry" of province *i* in the *t* year. Relevant data were obtained from the Wind database with some data loss, which are "health, social security and social welfare industry" and "public administration and social organizations" after 2012. Therefore, data gaps could be narrowed by checking the "Statistical Yearbook of the Chinese Investment in Fixed Assets". Step 2: We divided the "newly-added fixed assets by industry" into economic infrastructure and social infrastructure according to the industry classification mentioned above. Step 3: Considering the study by Jin [64] in the given year of 1997, we needed to convert our "newly-added fixed assets" of economic infrastructure and social infrastructure by the "price indices of investment in fixed assets" of province *i* in the *t* year at the 1997 constant price. It is worth noting that the data of "price indices of investment in fixed assets" were obtained from the Wind database, and the national index was used to make up for the lack of data on Guangdong and Hainan in some years. Step 4: From the definition of Equation (3), $K_{i,t}$ refers to the sum of economic infrastructure capital stock and social infrastructure capital stock. We then regarded the 2007 data in Table 5 of Jin [64] as the base period values of the "economic infrastructure capital stock" and "social infrastructure capital stock" of province *i*, with a list of initial annual values in 1997, 2000, 2004, 2008, and 2012. Therefore, the value at the beginning of the year in 2008 was adopted as the value at the end of the year in 2007 from Jin [64]. Step 5: According to the depreciation rate $\delta$ given by Jin [64], the comprehensive depreciation rate of economic infrastructure and social infrastructure was set as 9.21% and 8.51%, respectively. Step 6: Although it fails to assess the proportion of local government holdings in infrastructure from China's official statistical yearbooks, the information on "completion of urban fixed assets investment: central projects" and "com-

pletion of urban fixed assets investment: local projects" in the nine industries mentioned above from 2009 to 2017 is available from the Qianzhan database. To sum up these two sets of data, they provide access to information on the central projects and local projects related to infrastructure each year, which can help to divide the local projects to obtain the "proportion of local projects in infrastructure" during the sample period. Consequently, the annual average of the "proportion of local projects in infrastructure" was calculated to be 86.68%. Step 7: Some local projects are completed by private investment. It is important to estimate the "proportion of private investment in infrastructure" in this step. According to the data on "private fixed asset investment" and "completion of fixed assets investment" of the nine industries from 2012 to 2017, it was easy to obtain the "private fixed asset investment" and "completion of fixed assets investment" in infrastructure by summing up the year-end value of the nine industries. Then, by dividing these two data sets, we could obtain information on the "proportion of private investment in infrastructure" from 2012 to 2017. Finally, it was estimated that the annual average of "proportion of private investment in infrastructure" was 29.17%. The proportion of local government holdings in infrastructure varied to 86.68% × (1 − 29.17%) = 61.395%.

**Appendix E**

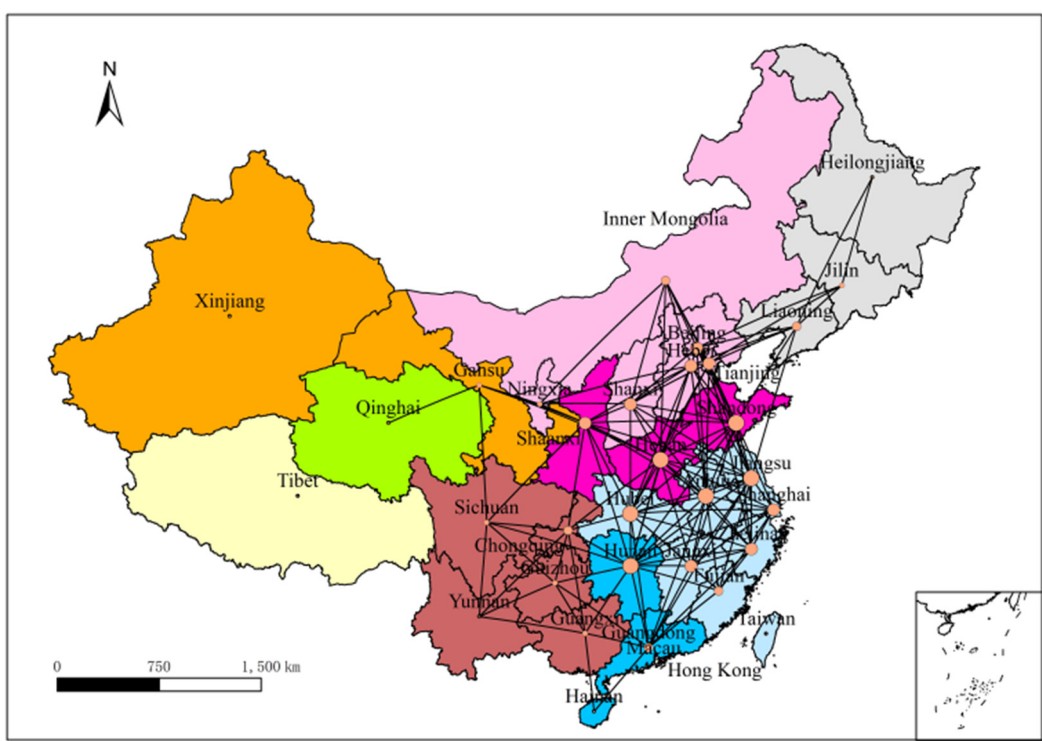

**Figure A1.** Spatial characteristics of China's local government debt risks in 2010.

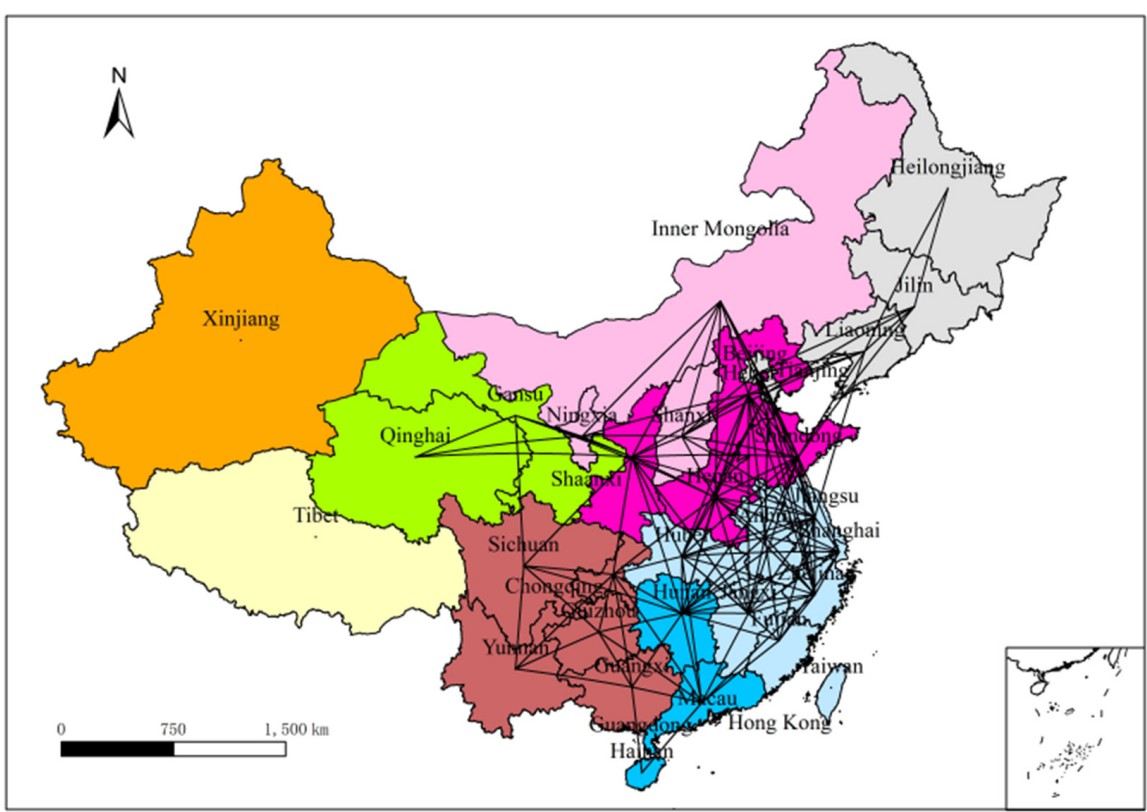

**Figure A2.** Spatial characteristics of China's local government debt risks in 2011.

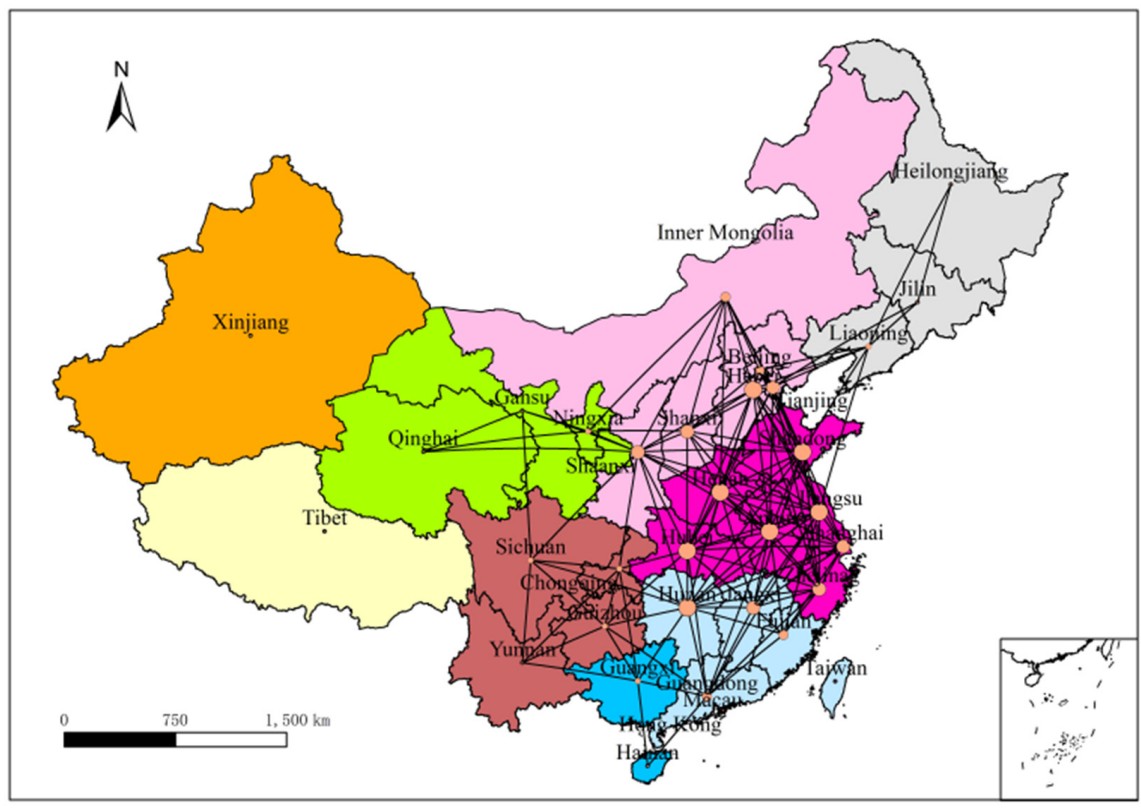

**Figure A3.** Spatial characteristics of China's local government debt risks in 2012.

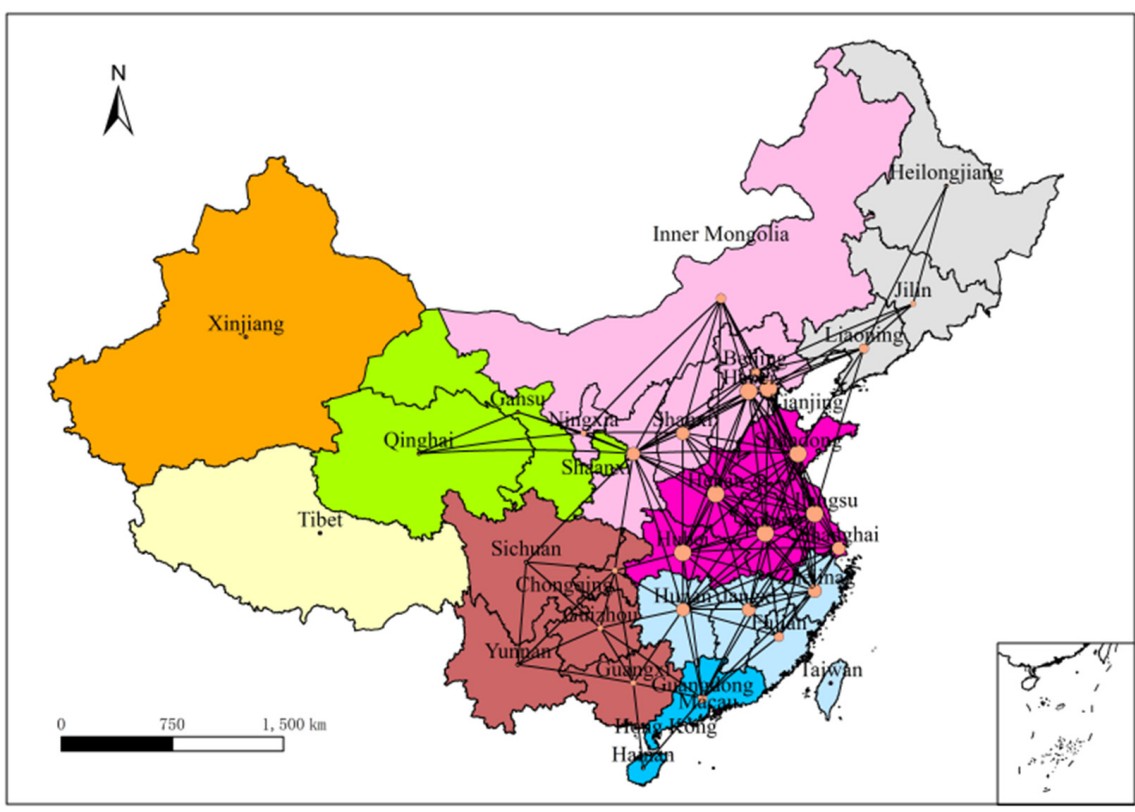

**Figure A4.** Spatial characteristics of China's local government debt risks in 2013.

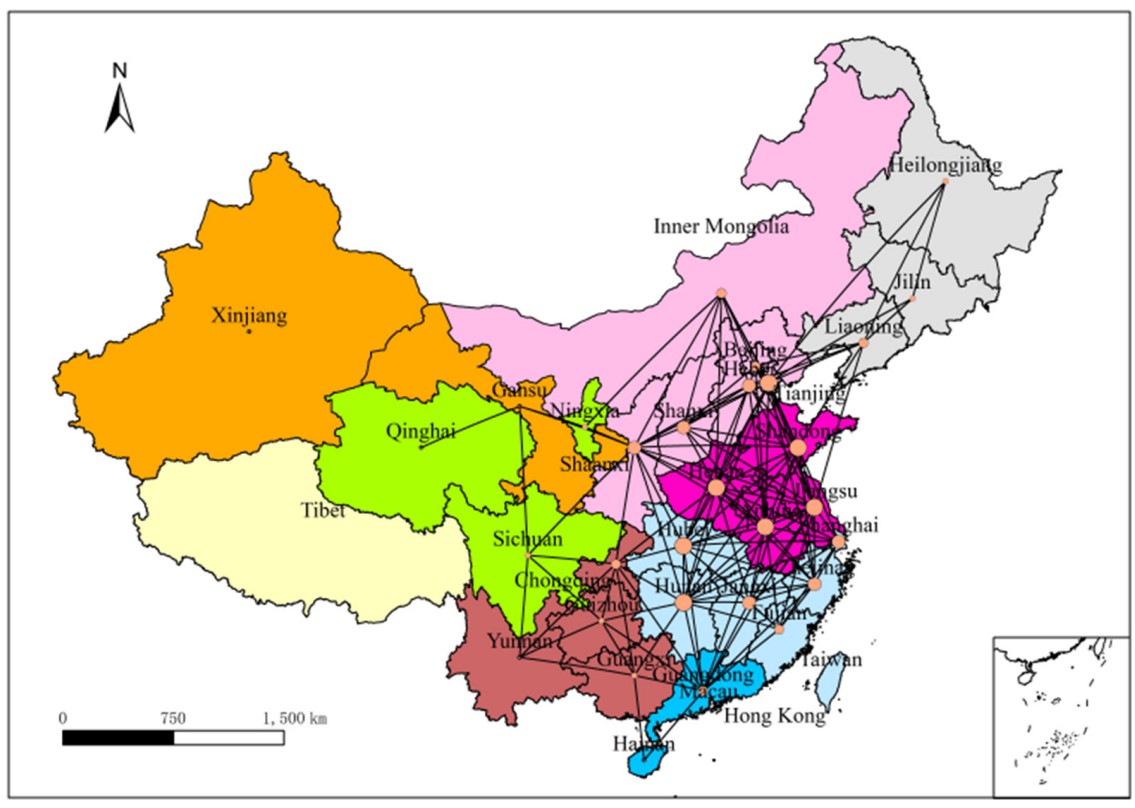

**Figure A5.** Spatial characteristics of China's local government debt risks in 2014.

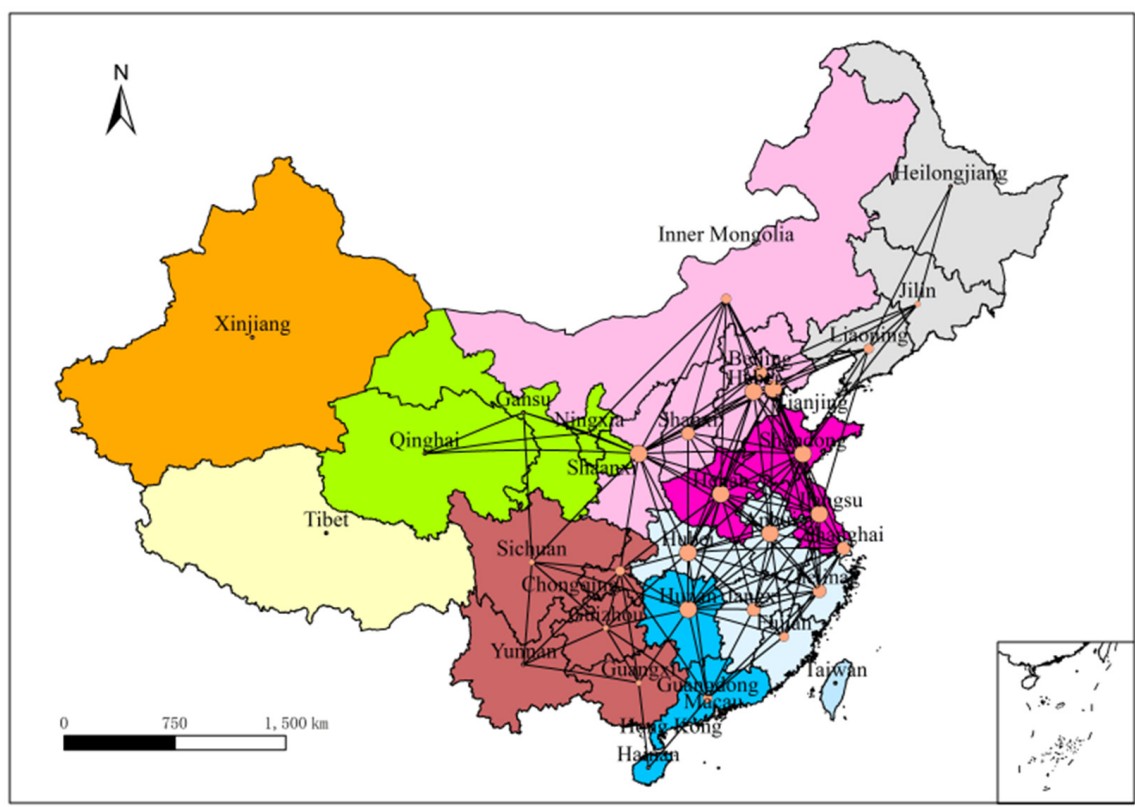

**Figure A6.** Spatial characteristics of China's local government debt risks in 2015.

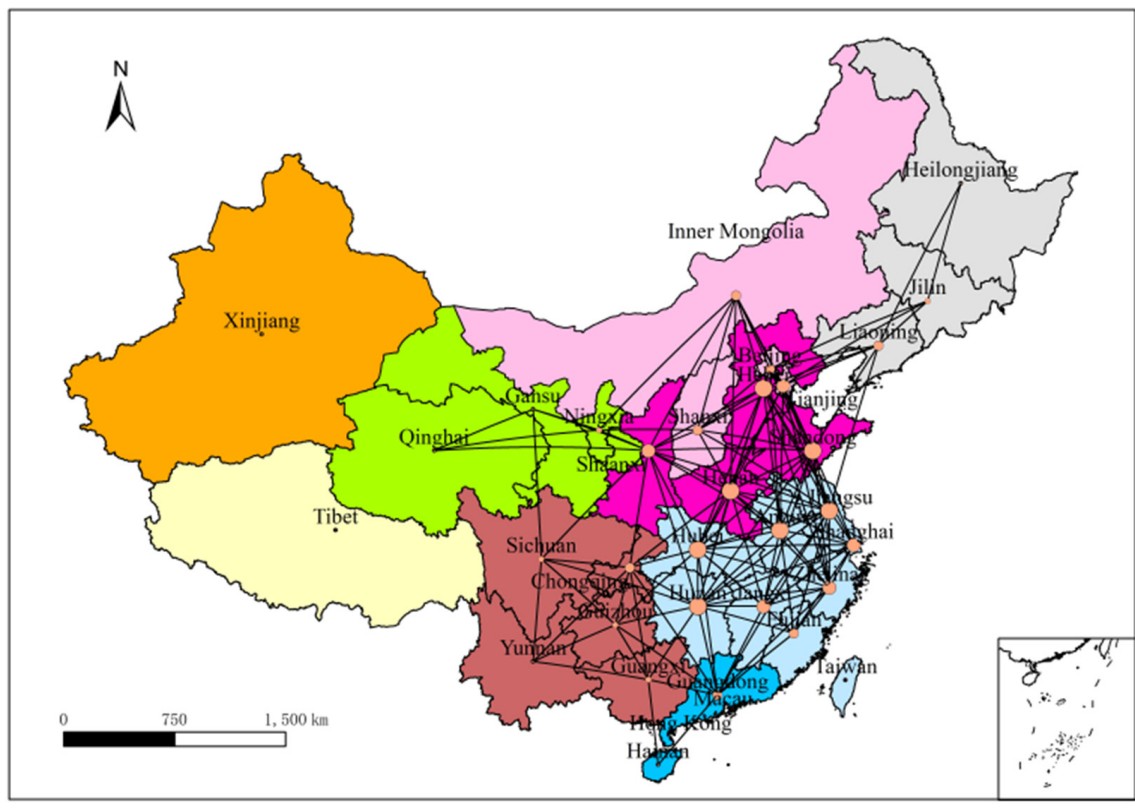

**Figure A7.** Spatial characteristics of China's local government debt risks in 2016.

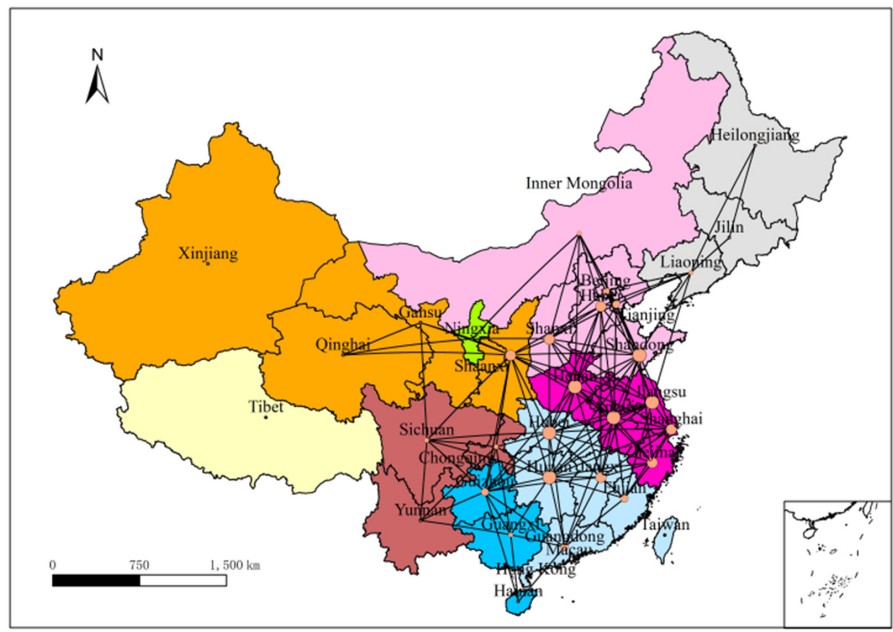

**Figure A8.** Spatial characteristics of China's local government debt risks in 2017.

## Appendix F

**Table A1.** Significance test for parameters in Table 2.

| Parameter | Year | Random Seed | Perms | *p*-Value | *p* > = Diff | *p* < = Diff |
|---|---|---|---|---|---|---|
| EI | 2009 | 16,629 | 5000 | 0.000 | 1.000 | 0.000 |
| | 2010 | 29,321 | | 0.000 | 1.000 | 0.000 |
| | 2011 | 4980 | | 0.000 | 1.000 | 0.000 |
| | 2012 | 1122 | | 0.000 | 1.000 | 0.000 |
| | 2013 | 19,599 | | 0.000 | 1.000 | 0.000 |
| | 2014 | 14,526 | | 0.000 | 1.000 | 0.000 |
| | 2015 | 26,602 | | 0.000 | 1.000 | 0.000 |
| | 2016 | 19,160 | | 0.000 | 1.000 | 0.000 |
| | 2017 | 30,421 | | 0.000 | 1.000 | 0.000 |
| | 2018 | 25,076 | | 0.000 | 1.000 | 0.000 |
| IL | 2009 | 16,629 | 5000 | 0.000 | 0.000 | 1.000 |
| | 2010 | 29,321 | | 0.000 | 0.000 | 1.000 |
| | 2011 | 4980 | | 0.000 | 0.000 | 1.000 |
| | 2012 | 1122 | | 0.000 | 0.000 | 1.000 |
| | 2013 | 19,599 | | 0.000 | 0.000 | 1.000 |
| | 2014 | 14,526 | | 0.000 | 0.000 | 1.000 |
| | 2015 | 26,602 | | 0.000 | 0.000 | 1.000 |
| | 2016 | 19,160 | | 0.000 | 0.000 | 1.000 |
| | 2017 | 30,421 | | 0.000 | 0.000 | 1.000 |
| | 2018 | 25,076 | | 0.000 | 0.000 | 1.000 |
| EL | 2009 | 16,629 | 5000 | 0.000 | 1.000 | 0.000 |
| | 2010 | 29,321 | | 0.000 | 1.000 | 0.000 |
| | 2011 | 4980 | | 0.000 | 1.000 | 0.000 |
| | 2012 | 1122 | | 0.000 | 1.000 | 0.000 |
| | 2013 | 19,599 | | 0.000 | 1.000 | 0.000 |
| | 2014 | 14,526 | | 0.000 | 1.000 | 0.000 |
| | 2015 | 26,602 | | 0.000 | 1.000 | 0.000 |
| | 2016 | 19,160 | | 0.000 | 1.000 | 0.000 |
| | 2017 | 30,421 | | 0.000 | 1.000 | 0.000 |
| | 2018 | 25,076 | | 0.000 | 1.000 | 0.000 |

**Table A1.** *Cont.*

| Parameter | Year | Random Seed | Perms | *p*-Value | *p* > = Diff | *p* < = Diff |
|---|---|---|---|---|---|---|
| Connectedness | 2009 | 16,629 | | 0.000 | 1.000 | 0.000 |
| | 2010 | 29,321 | | 0.000 | 1.000 | 0.000 |
| | 2011 | 4980 | | 0.000 | 1.000 | 0.000 |
| | 2012 | 1122 | | 0.000 | 1.000 | 0.000 |
| | 2013 | 19,599 | 5000 | 0.000 | 1.000 | 0.000 |
| | 2014 | 14,526 | | 0.000 | 1.000 | 0.000 |
| | 2015 | 26,602 | | 0.000 | 1.000 | 0.000 |
| | 2016 | 19,160 | | 0.000 | 1.000 | 0.000 |
| | 2017 | 30,421 | | 0.000 | 1.000 | 0.000 |
| | 2018 | 25,076 | | 0.000 | 1.000 | 0.000 |
| Least Upper Boundedness | 2009 | 16,629 | | 0.000 | 1.000 | 0.000 |
| | 2010 | 29,321 | | 0.000 | 1.000 | 0.000 |
| | 2011 | 4980 | | 0.000 | 1.000 | 0.000 |
| | 2012 | 1122 | | 0.000 | 1.000 | 0.000 |
| | 2013 | 19,599 | 5000 | 0.000 | 1.000 | 0.000 |
| | 2014 | 14,526 | | 0.000 | 1.000 | 0.000 |
| | 2015 | 26,602 | | 0.000 | 1.000 | 0.000 |
| | 2016 | 19,160 | | 0.000 | 1.000 | 0.000 |
| | 2017 | 30,421 | | 0.000 | 1.000 | 0.000 |
| | 2018 | 25,076 | | 0.000 | 1.000 | 0.000 |
| Characteristic Path Length | 2009 | 16,629 | | 0.000 | 0.000 | 1.000 |
| | 2010 | 29,321 | | 0.000 | 0.000 | 1.000 |
| | 2011 | 4980 | | 0.000 | 0.000 | 1.000 |
| | 2012 | 1122 | | 0.000 | 0.000 | 1.000 |
| | 2013 | 19,599 | 5000 | 0.000 | 0.000 | 1.000 |
| | 2014 | 14,526 | | 0.000 | 0.000 | 1.000 |
| | 2015 | 26,602 | | 0.000 | 0.000 | 1.000 |
| | 2016 | 19,160 | | 0.000 | 0.000 | 1.000 |
| | 2017 | 30,421 | | 0.000 | 0.000 | 1.000 |
| | 2018 | 25,076 | | 0.000 | 0.000 | 1.000 |
| Clustering Coefficient | 2009 | 16,629 | | 0.000 | 1.000 | 0.000 |
| | 2010 | 29,321 | | 0.000 | 1.000 | 0.000 |
| | 2011 | 4980 | | 0.000 | 1.000 | 0.000 |
| | 2012 | 1122 | | 0.000 | 1.000 | 0.000 |
| | 2013 | 19,599 | 5000 | 0.000 | 1.000 | 0.000 |
| | 2014 | 14,526 | | 0.000 | 1.000 | 0.000 |
| | 2015 | 26,602 | | 0.000 | 1.000 | 0.000 |
| | 2016 | 19,160 | | 0.000 | 1.000 | 0.000 |
| | 2017 | 30,421 | | 0.000 | 1.000 | 0.000 |
| | 2018 | 25,076 | | 0.000 | 1.000 | 0.000 |
| Transitivity | 2009 | 16,629 | | 0.000 | 1.000 | 0.000 |
| | 2010 | 29,321 | | 0.000 | 1.000 | 0.000 |
| | 2011 | 4980 | | 0.000 | 1.000 | 0.000 |
| | 2012 | 1122 | | 0.000 | 1.000 | 0.000 |
| | 2013 | 19,599 | 5000 | 0.000 | 1.000 | 0.000 |
| | 2014 | 14,526 | | 0.000 | 1.000 | 0.000 |
| | 2015 | 26,602 | | 0.000 | 1.000 | 0.000 |
| | 2016 | 19,160 | | 0.000 | 1.000 | 0.000 |
| | 2017 | 30,421 | | 0.000 | 1.000 | 0.000 |
| | 2018 | 25,076 | | 0.000 | 1.000 | 0.000 |

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
