# Peer review of "Research on Spatial Correlation Characteristics and Their Spatial Spillover Effect of Local Government Debt Risks in China"

_sustainability, doi:10.3390/su13052687_

Round 1

Reviewer 1 Report

Dear Authors,

The revised version is satisfactory.

Author Response

Thanks for Reviewer's valuable work. Your support and encouragement are very important for us.

Best wishes,

Sincerely yours,

Xing Li, Xiangyu Ge, Wei Fan, Hao Zheng

Reviewer 2 Report

I reviewed this paper a second or third time and now I observed huge progress and I accept implemented changes.
I think there are some technical errors in the References section, which can be solved later by the technical editors.

Reviewer 3 Report

Dear Authors,

You have made efforts to improve your paper, especially in presenting the general context of local debt issue in China.

Your analysis seems to be well rooted and accounts for local specifics and local factors of influence. The methods you used seem to correspond to the hypotheses you have presented.

Here, it would help if you state who are the main underwriters of China's local debt issues, especially of bonds. Are they rational investors, which account for economic and financial risks of the issuing entity? Or they also suffer from limitations and constraints which favours investments in such assets? Is local government debt quality and volume appraised by some independent financial institution or agency?

It seems English degraded a bit, probably from adding a lot of new affirmations. So, you should improve English all across the paper.

You still have not made any significant comparisons with previous similar research, made for other countries. You should improve this part.

I think that if you respond to the suggestions I have made your paper can be a candidate for publishing in this journal.

Best regards

Author Response

This manuscript is a resubmission of an earlier submission. The following is a list of the peer review reports and author responses from that submission.

Round 1

Reviewer 1 Report

Dear Authors,

The submitted manuscript is interesting. The structure of the manuscript is logical and the methodology is appropriate. I suggest some improvements.

I would replace the "Introduction and Literature Review" section with "Introduction" only.

Figures 1a to 1i can be transferred to the attachment.

I would extend the section "Conclusions and Policy Recommendations" to include a discussion. This element is missing from the manuscript.

In the "Conclusions and Policy Recommendations" section, reference can be made to recommendations not only for the policy but also for the effects on residents.

Author Response

Please see the attachment(pdf)

Reviewer 2 Report

Dear Authors,

I think that your title is too long. Please think about a shorter title maximum to lines of text, not four as it is now.

In my opinion, the introduction and literature review should be separate sections.

Lines 29-45 have no literature background so they can be named as author opinion, which is not proper for the very first paragraph of the introduction section.

Line 36 inappropriate citation format for Jiang 2018 work. 

If this section is a literature review, what method of the literature review was applied, and why the whole article is based on the 46 sources (articles, books reports). These reports are only focused on China only 98% of the cited authors are Chinese scholars, this allows me to conclude that this paper has nothing to offer the broader audience and it is too China-centric. This paper has no relation to the other countries, no similarities for other parts of the world are indicated. There is no possibility to extend proposed findings to the other countries in Asia or South America. This makes the paper not interesting for readers outside of China.

line 29: Why do you think the sentence "Nowadays, the growth of the global economy has the tendency to slow down" is true? what is the source of this idea?

Lines 31-32: you wrote: "In China, the slowdown of economic growth is usually accompanied by a large-scale expansion of local government debt." does it happen frequently or in some periods of time that you observe it often or "usually"?

Lines 32-34 - there is no citation supporting these numbers.

There is no research gap identified in the introduction and literature review section. There is no aim (goal) of the paper defined in the introduction section.

What is the difference between sigmas used in equation 3 and equation 4? why all the used symbols are not explained under the equations?

Why you present all figures (Fig 10.a-j) for the years 2009 -2018 instead of choosing just 2 of them and move the rest to appendices? Only two significant and indicating visible differences between them are useful. For a reader, there is no clear difference between these figures (10.a-j) 

The proposed models have no significant value (Table 2), so your research has not proved any hypothesis provided in the 2nd section. The logic value of v(p) =0 if "the expected values refer to the values randomly distributed within or between the groups. Then, the difference is defined by subtracting the expected from the observed. Specifically, if the difference is larger than or equal to 0, P>= Diff; otherwise, P<= Diff. " as you wrote in the lines 702-705. 

There is no proof for the linearity of the correlation you claim that exists in your research, if there is a linear model why you don't present the normal distribution of the residuals? Where is a Durbin-Watson test to prove if the variables present no colinearity to each other?

There is no discussion of the proposed models, do they fit the other presented in the literature models.

The hypotheses were not proved even partially even you claim that you proved the first or second part of your hypothesis.

There is no table with the correlations presented between variables. It is impossible to repeat your calculations since the raw data are not accessible in any repository.

The provided statistical calculations are misleading, there is no proof for any part of the stated hypotheses; The proposed models are extremely week R2 = 0,266 in Table 3 and R2 = 0,355 in Table 5. Why your R2 values are different for the formulated model (in Table 3) and Validation table 5?
There is no linearity between models, therefore the relations cannot be represented by lines. The proposed network model (Figure 10.a-j) is just the result of Arc Map 10.5 software and are not the result of the calculations. The model of nodes and edges is not suitable for this type of analysis.

As the statistician, I expect to see checking the linear model assumptions:

  1. the significance of linear regression;
  2. the importance of partial regression coefficients;
  3. no collinearity (redundancy) between independent variables;
  4. homoscedasticity assumption, which means that the variance of the random component (εi) is the same for all observations;
  5. no residual autocorrelation;
  6. normal residual distribution;
  7. the random term εi has the expected value equal to 0 (if linear model).

If only one of the mentioned points is not fulfilled we cannot think even about linearity. If you are sure of your calculations please provide proves in the appendices section.

Author Response

Please see the attachment(pdf)

Reviewer 3 Report

Introduction and Literature Review should be split in two different sections.

The Introduction should highlight the relevance of the topic, the novelty of the results, the importance of policy implications, the sample’s choice, the methodology’s appropriateness, the data used, the contribution to the literature, and the limitations of the study.

The theoretical analysis of fiscal sustainability is far from the standard analysis present in the literature. 

Literature review is partial and incomplete, and some recent and relevant contributions should be cited and discussed: i.e., 10.1002/ijfe.2184; 10.1016/j.jeca.2019.e00127; 10.1186/s40008-019-0151-5; 10.1007/s11293-018-9588-4; 10.1002/soej.12269.

Dataset should be discussed more in detail.

Comparisons with previous studies are absent.

Policy implications are weak.

The English needs a proof-reading by a native speaker.

The paper does not fit with the journal's scope very well.

Author Response

Please see the attachment(pdf)

Reviewer 4 Report

Hello, Dear Authors,

First of all, congratulations for the idea and your work. The Chinese case is interesting since it seems to be a mixture between the constraints and specifics of a centralised economy and many elements of the free market economy, with local fiscal decentralisation, competition for financial resources, local initiatives and the manifestation of selection criteria based on efficiency and environmental impact.

It seems you have put some effort into this paper and you have obtained a decent final output.

First of all, I would recommend changing the title of the paper "The Spatial Correlation Characteristics and Spillover Effect of Local Government Debt Risks in China: An Empirical Explanation Different from the Sustainability of Local Government Debt". The title is too long and unclear in its second part. We realise that you are aiming to publish in Sustainability, yet sometimes it seems a little too much when over-using the word sustainability, especially when you do not address per se the sustainability of local government debt.

I suggest you include a short presentation of China's current status of local Governments attributes and autonomy so the readers can better understand the general context of your paper.

I would recommend you including some more relevant papers in your references, since now the vast majority is represented by Chinese authors.

I would recommend at least

"Spillover effects of taxes on government debt: a spatial panel approach" by Kopczewska, K.; Kudla, J; Walczyk, K.; Kruszewski, R.; Kocia, A., Policy Studies, 2016.

Also,

"Race to the debt trap? - Spatial econometric evidence on debt in German municipalities", Borck, R.; Fossen, FM.; Freier, R.; Martin, T., REGIONAL SCIENCE AND URBAN ECONOMICS, 2015   "Soft budget constraints and strategic interactions in subnational borrowing: Evidence from the German States, 1975-2005", Baskaran, T., JOURNAL OF URBAN ECONOMICS, 2011.   These references and probably others alike are also necessary to improve your Conclusions section. You have to state how your research positions itself in comparison with similar studies.   I consider that after you address my suggestions from above the paper can be a good candidate for publication.  

Best Regards    

Author Response

Please see the attachment(pdf)

Round 2

Reviewer 2 Report

Dear Authors,

With pleasure, I read your improved paper. I appreciate your efforts and the changes implemented in this paper. 

However, I found small mistakes, as follow:

1. you are referring to Harris and Piwowar [27] 
but actually, it should be Harris et al. because in your literature there is: Harries, Lawrence E.; Michael, S., Piwowar. Secondary Trading Costs in the Municipal Bond Market. Journal of Finance. 2006, 61, 1361-1397

2. Please also consider these papers to make your paper more international and open new future research avenues: 

Zema, T., Sulich, A.: Relations in The Interorganizational Networks. (2019). https://doi.org/10.23918/ijsses.v6i1p111.

Rutkowska, M. et al.: Debt Capacity of Nonprofit Entities: Jobs on the Green Economy Background Case. In: Gavurová, B. and Šoltés, M. (eds.) Proceedings of the 2 nd International Scientific Conference Central European Conference in Finance and Economics (CEFE2017). pp. 687–697; Technical University of Košice, Košice (2017).

3. Can you please start footnote 7 with the explanation of what is calculated: 
Please begin the footnote as  "the proportion of local government holdings in infrastructure estimated as" and then calculations

4. Footnote 10 should be considered to be replaced by the normal citation from the website:  国证指数网-指数编制方案, http://www.cnindex.com.cn/module/pdf-detail.html?pdf=/docs/gz_921112.pdf&name=江苏省地债, last accessed 2021/01/29.
But please translate from the Chinese language and provide the Institution or Author (Surname and Name) 

My recommendations and opinion (you don't have to reply to this part)

I use Mendeley by Elsevier (both Desktop and Website importer) to use the Sustainability Journal style in the word written article; this makes writing much more simple. 

With pleasure, I reviewed your paper. I am waiting for the last suggested changes to be implemented and you are on the best way to publish your paper

Reviewer 3 Report

Accept

Author Response

We appreciate the reviewer’s acknowledgement of our work.